

# Guidance on evaluating parametric model uncertainty at decision-relevant scales

Jared D. Smith[1], Laurence Lin[2], Julianne D. Quinn[1], and Lawrence E. Band[1,2]

[1]Department of Engineering Systems and Environment, University of Virginia, Charlottesville, VA, USA
[2]Department of Environmental Sciences, University of Virginia, Charlottesville, VA, USA

**Correspondence:** Jared D. Smith (js4yd@virginia.edu; jared.d.smith485@gmail.com)

**Abstract.** Spatially distributed hydrologic models are commonly employed to optimize the locations of engineering control measures across a watershed. Yet, parameter screening exercises that aim to reduce the dimensionality of the calibration search space are typically completed only for gauged locations, like the watershed outlet, and use screening metrics that are relevant to calibration instead of explicitly describing decision objectives. Identifying parameters that control physical processes in ungauged locations that affect decision objectives should lead to a better understanding of control measure effectiveness. This paper provides guidance on evaluating model parameter uncertainty at the spatial scales and flow magnitudes of interest for such decision-making problems. We use global sensitivity analysis to screen parameters for model calibration, and to subsequently evaluate the appropriateness of using parameter multipliers to further reduce dimensionality. We evaluate six sensitivity metrics that align with four decision objectives; two metrics consider model residual error that would be considered in spatial optimizations of engineering designs. We compare the resulting parameter selection for the basin outlet and each hillslope. We also compare basin outlet results to those obtained by four calibration-relevant metrics. These methods were applied to a RHESSys ecohydrological model of an exurban forested watershed near Baltimore, MD, USA. Results show that 1) the set of parameters selected by calibration-relevant metrics does not include parameters that control decision-relevant high and low streamflows, 2) evaluating sensitivity metrics at only the basin outlet does not capture many parameters that control streamflows in hillslopes, and 3) for some parameter multipliers, calibration of just one of the parameters being adjusted may be the preferred approach for reducing dimensionality. Thus, we recommend that parameter screening exercises use decision-relevant metrics that are evaluated at the spatial scales appropriate to decision making. While including more parameters in calibration will exacerbate equifinality, the resulting parametric uncertainty should be important to consider in discovering control measures that are robust to it.

## 1 Introduction

Spatially distributed hydrologic models are commonly employed to optimize the locations of engineering control measures across a watershed. Accurate simulations of streamflows and nutrient fluxes in ungauged locations are desired to estimate the impact of control measures on multiple objective functions (e.g., Maringanti et al., 2009). However, these models can have many hundreds of parameters that cannot feasibly be measured throughout the watershed, and some parameters are



not observable even with state-of-the-art equipment. Thus, parameter estimation through calibration is required. To reduce
the dimensionality of the parameter search space, parameter screening exercises are usually completed via sensitivity analysis.
Reviews of sensitivity analysis methods and guides specifically applied to spatially distributed environmental and earth systems
models have recently been provided by many authors (Pianosi et al., 2016; Razavi and Gupta, 2015; Koo et al., 2020b; Lilburne
and Tarantola, 2009). These reviews and other studies have documented the critical need to answer "What is the intended
definition for sensitivity in the current context?" (Razavi and Gupta, 2015) at the outset of a study. For studies that aim to use
the resulting model to spatially optimize decisions, the definition of sensitivity metrics should match the decision objectives.
However, Razavi et al. (2021) note that "Studies with formal [sensitivity analysis] methods often tend to answer different (often
more sophisticated) questions [than] those related to specific quantities of interest that decision makers care most about."

The large majority of studies use sensitivity metrics that are relevant to model calibration objectives (aiming for the best
model fit), rather than explicitly focusing on how the model will be used to evaluate decision-making objectives. Common
calibration performance measures emphasize specific features of a hydrological time series, and some measures like the Nash-
Sutcliffe Efficiency lump several features together (Gupta et al., 2009). Matching a hydrological time series well for all flows
might be important for ecological investigations (Poff et al., 1997), but may complicate the analysis for the purpose of engineer-
ing control measures that are mainly concerned with controlling extremes. Furthermore, calibration data are often limited to
few gauged locations or only the watershed outlet, so sensitivity analyses based on calibration metrics only screen parameters
that influence flows at gauged locations (e.g., van Griensven et al., 2006). Yet locations of engineering control measures will
be affected by the parameters that control physical processes in their local area, which may be different than the parameters
that have the largest signals in model outputs at the watershed outlet or other gauged locations (e.g., Golden and Hoghooghi,
2018). This would suggest there is equifinality of model parameter sets (e.g., Beven and Freer, 2001), which simulate similar
model output values at gauged locations, yet simulate different values elsewhere across the watershed.

The combination of these factors could have proximate consequences on siting and sizing engineering controls if equifinal
parameter sets for the watershed outlet 1) suggest different optimal sites and/or sizes due to the resulting uncertainty in model
outputs across the watershed, or 2) do not consider all of the relevant parametric uncertainties across the watershed. This paper
provides guidance on evaluating parametric model uncertainty at the spatial scales and flow magnitudes of interest for such
decision-making problems as opposed to using a single location and metrics of interest for calibration. We use three sensitivity
metrics to capture differences in parameters that control physical processes that generate low flows, flood flows, and all other
flows as in Ranatunga et al. (2016), but extend the analysis to consider the decision-relevant implications for calibration to
ensure robust engineering design. Because stochastic models are required for risk-based decision making (Vogel, 2017), we
use another three sensitivity metrics to compare parameters screened for calibration using deterministic mean values to those
screened using upper and lower quantiles of model residual error. We refer to these six metrics as decision-relevant sensitivity
metrics. We compare the parameters screened from these metrics to those screened from four commonly employed calibration
metrics. Finally, we illustrate the value of spatially distributed sensitivity analysis by comparing parameter selections for
the watershed outlet with parameter selections for each hillslope outlet (i.e., the water, nutrients, etc. contributed to a sub-
watershed outlet by a hillslope). The goal is to discover to which parameters the decision objectives are most sensitive across





the watershed. With these approaches, this paper contributes to a limited literature on sensitivity analysis to inform parameter
screening of spatially distributed models that will be used in engineering decision making.

We employ the RHESSys ecohydrological model for this study (Tague and Band, 2004). The results we obtain from the
comprehensive sensitivity analysis of all non-structural model parameters are used to provide general guidelines for spatially
distributed models, with some specific recommendations for RHESSys users. Our results are also used to inform prioritization
of data collection efforts for the study watershed based on those parameters that spatially have the greatest impact on sensitivity
metrics. We then consider parameter multipliers as a further dimensionality reduction technique that is commonly employed
for calibrations of spatially distributed models (e.g., soil and vegetation sensitivity parameters in RHESSys (Choate et al.,
2020), soil parameter ratios in a SAC-SMA model (Fares et al., 2014), climatic multipliers in a SWAT model (Leta et al.,
2015), and many others (Pokhrel et al., 2008; Bandaragoda et al., 2004; Canfield and Lopes, 2004)). The multiplier adjusts the
base values of parameters of a certain type (e.g., soil hydraulic conductivity) and only the multiplier is calibrated. This method
can be useful to reduce the number of calibration parameters while capturing spatial trends, but there are known limitations to
the methodology (Pokhrel and Gupta, 2010). In particular, for a set of parameters with different magnitudes, a multiplier will
disproportionately adjust the mean and variance of parameters' distributions, and could lead to poor performance in ungauged
locations. This paper provides guidance on the use of multipliers by examining model sensitivity to individual parameters in
the set of adjusted parameters.

The remainder of the paper is structured as follows. Section 2 details the methods used to screen parameters and evaluate pa-
rameter multipliers using global sensitivity analysis, Section 3 describes the RHESSys model and its parameters considered for
this study, and Section 4 describes the study watershed. The subsequent sections present the results, discussion and concluding
thoughts.

## 2    Methods

### 2.1    Uncertainty Sources Considered for Sensitivity Analysis

Uncertainty sources in all environmental systems models include (e.g., Vrugt, 2016, Fig. 1): the model structure (e.g., selection
of process equations (Mai et al., 2020) or grid cell resolution (Melsen et al., 2019; Zhu et al., 2019)), initial condition values
(e.g., groundwater and soil moisture storage volumes (Kim et al., 2018)), model parameter values (Beven and Freer, 2001),
and input data (e.g., precipitation and temperature in Shields and Tague (2012)). If employing a stochastic modeling approach
to these deterministic models (Farmer and Vogel, 2016), as is recommended for risk-based decision making (Vogel, 2017),
additional uncertainty sources include the choice of error model shape (e.g., lognormal) (Smith et al., 2015), the error model
parameter values, and the observation data that are used to compute the residual errors (McMillan et al., 2018). Each of these
uncertainty sources could be considered in a sensitivity analysis.

In this paper, the sensitivity analyses consider parametric uncertainty for a fixed RHESSys model structure and input data
time series (described in Section 3). The analysis corresponds to the mean of a stochastic process that is conditional on this
assumed model structure. We did not consider stochastic methods because sensitivity will be evaluated in ungauged locations





where no data are available to inform an error model. However, we do evaluate the impact of considering model error for the regression model that was used to estimate total nitrogen concentrations from RHESSys outputs, as described in Section 2.2.

We address uncertainty in the initial conditions for RHESSys by employing a five year spin-up period before using simulated outputs for analysis. After five years, the water storage volume (saturation deficit) averaged over the watershed maintained a nearly stationary mean value for each of the evaluated parameter sets (supplementary material item S3).

## 2.2 Decision-Relevant Sensitivity Metrics

In many hydrological studies, sensitivity analysis is used to understand how input parameters influence model performance

measures (Jackson et al., 2019), such as the Nash-Sutcliffe efficiency. Gupta and Razavi (2018) note that using such performance measures as sensitivity metrics amounts to a parameter identification study to discover which parameters may be adjusted to improve model fit. Evaluating performance measures for sensitivity metrics that describe specific features of the hydrological time series (Olden and Poff, 2003) should aid identification of parameters that control processes that generate those features. If sensitivity metrics correspond to decision objectives, then identified parameters directly relate to the objec-

tives of interest. The performance measure, decision objectives, and resulting decision-relevant sensitivity metrics are described in this section.

The sum of absolute error (SAE) is the performance measure used for decision-relevant sensitivity metrics. The SAE for the basin outlet was computed relative to the observed data, and for hillslopes was computed relative to the median across all simulations on days that data were sampled at the basin outlet, as shown in Equations 1 and 2:

$$Basin : SAE = \sum_{t=1}^{T} |\boldsymbol{Q_{sim}}[t] - \boldsymbol{Q_{obs}}[t]| \quad (1)$$

$$Hillslope : SAE = \sum_{t=1}^{T} |\boldsymbol{Q_{sim}}[t] - \mathrm{med}(\mathbf{Q_{sim}}[:,t])| \quad (2)$$

where $T$ is the total number of time series data points for the sensitivity metric, $\boldsymbol{Q_{sim}}$ is the time series of the simulated quantity (e.g., streamflow), $\boldsymbol{Q_{obs}}$ is the vector of the observed quantity, and $\mathrm{med}(\mathbf{Q_{sim}}[:,t])$ is the median simulated quantity at time $t$ over all of the model runs completed for sensitivity analysis, as stored in matrix $\mathbf{Q_{sim}}$.

We consider water quantity and quality objectives as they are among the most common for hydrological modeling studies. We evaluate three streamflow sensitivity metrics relevant to flooding, low flow, and reservoir water supply objectives, respectively. These mutually exclusive objectives are respectively quantified as 1) flows greater than the historical $95^{th}$ percentile, 2) flows less than the historical $5^{th}$ percentile, and 3) flows between the historical $5^{th}$ and $95^{th}$ percentiles. The percentiles are estimated based on the calibration data (described in Section 4). Variability in the resulting sensitivity metrics and screened parameters

would be a function of the physical processes that generate these flows. The dates corresponding to flood flows provided a good sampling across all years of record. For low flows, most dates correspond to a drought in 2007. Therefore, using the historical $5^{th}$ percentile as a metric could capture decision-relevant low flows, but potentially be overly sensitive to one particular period of the record. We compared results obtained from using each water year's daily flows less than that year's $5^{th}$ percentile to results obtained from using the historical $5^{th}$ percentile. The parameters that would be selected for calibration were identical





for the example presented in this paper, so we display only the historical $5^{th}$ percentile results. The resulting sensitivity metrics
for these objectives compute the SAE only for the $T$ days on which the objectives are defined.

The water quality objective considers reducing the estimated daily total nitrogen (TN) concentration. As described in Section
3.1, we used a statistical method to estimate TN concentration as a function of streamflow. The water quality sensitivity metrics
corresponded to estimated quantiles from the regression error model: 1) $95^{th}$, 2) $5^{th}$, and 3) log-space mean (real space median)
on each of the days on which TN was sampled. Therefore, unlike the streamflow objectives, these objectives reveal variability
in the resulting sensitivity metrics as a function of uncertainty in the TN estimation method. The purpose of these metrics is to
test whether or not different parameters are screened for different error quantiles.

### 2.3 Calibration-Relevant Sensitivity Metrics

Four calibration performance measures are used as calibration-relevant sensitivity metrics: the Nash-Sutcliffe efficiency (NSE),
the NSE of log-space simulations (LNSE), the percent bias (pBias), and the log of the likelihood model that describes residual
errors for streamflow. These metrics can only be computed for gauged locations, which is the basin outlet in this study. The
first three metrics are defined in Equations 3 to 5

$$NSE = 1 - \frac{\sum_{t=1}^{T}(\boldsymbol{Q_{sim}}[t] - \boldsymbol{Q_{obs}}[t])^2}{\sum_{t=1}^{T}(\boldsymbol{Q_{obs}}[t] - \mathbb{E}[\boldsymbol{Q_{obs}}])^2} \quad (3)$$

$$LNSE = 1 - \frac{\sum_{t=1}^{T}(\ln[\boldsymbol{Q_{sim}}[t]] - \ln[\boldsymbol{Q_{obs}}[t]])^2}{\sum_{t=1}^{T}(\ln[\boldsymbol{Q_{obs}}[t]] - \mathbb{E}[\ln(\boldsymbol{Q_{obs}})])^2} \quad (4)$$

$$pBias = 100 \times \frac{\sum_{t=1}^{T}(\boldsymbol{Q_{sim}}[t] - \boldsymbol{Q_{obs}}[t])}{\sum_{t=1}^{T}\boldsymbol{Q_{obs}}[t]} \quad (5)$$

where ln is the natural logarithm, $\mathbb{E}$ is the expectation operator and other terms are as previously defined. The NSE is more
sensitive to peak flows due to the squaring of residual errors, so it is hypothesized that parameters screened by NSE will be
most similar to those screened by the flood flow decision objective, although there are known issues with using NSE as a peak
flow metric (e.g., Mizukami et al., 2019). The LNSE squares log-space residuals, so it assigns more equal weight to all flows;
however, it is common to use LNSE as a low flow calibration objective. The pBias considers the scaled raw error, so it should
assign the most equal weight to all flows.

We selected the likelihood model based on a need to fit a wide variety of residual distribution shapes that could result from
random sampling of hydrological model parameters in the sensitivity analysis. We selected the skew exponential power model
(Schoups and Vrugt, 2010), which is a generalized normal distribution. We used the implementation with two additional param-
eters that describe heteroskedasticity as a function of flow magnitude and a lag-1 autocorrelation, both of which are common
in hydrological studies. The probability density function and resulting log likelihood (LogL) have lengthy derivations provided





in (Schoups and Vrugt, 2010), as summarized in Appendix A with minor changes for our study. We used maximum likelihood estimation to obtain point estimates of the six likelihood model parameters, as described in supplementary information (item S0). We assume that this likelihood model would be maximized in calibration of the selected model parameters, so its selection

as a sensitivity metric directly represents the calibration objective function.

## 2.4 Morris Global Sensitivity Analysis

Sensitivity analysis methods can be local about a single point, or global to summarize the effects of parameters on model outputs across the specified parameter domain (e.g., Pianosi et al., 2016). A global method is implemented for this study because the goal is to screen parameters for use in model calibration. The Method of Morris (1991) derivative-based sensitivity

analysis is employed as a computationally fast method whose parameter rankings have been shown to be similar to more expensive variance-based analyses (Saltelli et al., 2010) for spatially distributed environmental models (Herman et al., 2013a).

  The Method of Morris is based on elementary effects (EEs) that approximate the first derivative of the sensitivity metric with respect to a change in a parameter value. EEs are computed by changing one parameter at a time along a trajectory, and comparing the change in sensitivity metric from one step in the trajectory to the next. The change is normalized by the relative

change in the parameter value (Eq. 7). Assuming that the $p^{th}$ parameter is changed on the $(s+1)^{th}$ step in the $j^{th}$ trajectory, the EE for parameter $p$ using the computed sensitivity metrics (SMs) (SAE, NSE, etc.) is computed as shown in Equation 6:

$$\mathbf{EE}[j,p] = \frac{\mathbf{SM}[j,s+1] - \mathbf{SM}[j,s]}{\Delta_{s+1,s,p}} \tag{6}$$

$$\Delta_{s+1,s,p} = \frac{\mathbf{X}[j,s+1,p] - \mathbf{X}[j,s,p]}{|\max(\mathbf{X}[:,:,p]) - \min(\mathbf{X}[:,:,p])|} \tag{7}$$

where $\mathbf{EE}$ is the elementary effect matrix consisting of one row per trajectory and one column per parameter, $\Delta_{s+1,s,p}$ is the

change in the value of the parameter as a fraction of the selected parameter range, and $\mathbf{X}$ is the matrix of parameter values. EEs for each parameter are typically computed in tens to hundreds of locations in the parameter domain, and are then summarized to evaluate global parameter importance. The mean absolute value of the EEs computed over all of the $r$ locations (one for each trajectory) is the summary statistic used to rank model sensitivity to each parameter, as recommended by Campolongo et al. (2007). The sample estimator is provided in Equation 8:

$$\hat{\mu_p^*} = \frac{1}{r} \sum_{j=1}^{r} |\mathbf{EE}[j,p]|. \tag{8}$$

  We used 40 trajectories that were initialized by a Latin hypercube sample, and used the R sensitivity package (Iooss et al., 2019) to generate sample points and compute EEs. Each parameter had 100 possible levels that were uniformly spaced across its specified range. Step changes in parameter values along the trajectory were set to 50 levels to allow for a uniform sampling distribution for each parameter within the specified bounds.



### 2.4.1 Elementary Effects for Parameters with Relational Constraints

RHESSys and many other environmental systems models have parameters that are structurally dependent (e.g., sand% + silt% + clay% = 100%, leaf area index for tree species A < tree species B). For such parameters, their effects on model outputs cannot be uniquely identified (Guillaume et al., 2019) using the independence assumptions required of most sensitivity analysis methods. While algorithms are readily available to sample from high-dimensional spaces with relational constraints among dimensions (e.g., Beal et al., 2014), research is needed to develop trajectory-based sensitivity analysis sampling designs that also obey the constraints while perturbing parameters with the same relative jump sizes as all unconstrained parameters (i.e., sensitivity is conditional on the perturbation scale, Haghnegahdar and Razavi, 2017). For this paper, we adjust the original trajectory steps to meet the constraints, and implement an EE aggregation method for parameters that were related by constraints.

We handled simplex constraints by: 1) computing the sum, $S$, of the original parameter values obtained from the Morris sampling method 2) computing the difference, $\delta$, between $S$ and the sum required by the constraint, and 3) evenly allocating $\delta$ to each of the summed parameters while ensuring that all parameters remained within or at their bounds. We handled inequality constraints, where one parameter must be less than another, by finding the parameter with the smaller lower bound and resampling its value to be between its lower bound and the value of the other parameter. This method relied upon strategic selection of lower and upper bounds for parameters that had to jointly satisfy many relational constraints. We updated the Morris trajectories with the resulting parameter values so that the chains were continuous.

As a result of these imperfect sampling methods, multiple parameter values may change in a single Morris step; thus, EEs for parameters with relational constraints may be biased relative to EEs for other parameters that were all adjusted one at a time. As a simple example, consider a system with output variable $Y = f(X_1, X_2)$, and constraint $X_1 < X_2$. If the step change in $X_2$ is such that the constraint is satisfied, then the EE would reflect only a change in $X_2$ on $Y$, as desired. If instead $X_1$ must be adjusted to satisfy the constraint, then the EE would reflect changing $X_1$, $X_2$, and the interaction of $X_1$ and $X_2$. An additional problem with the sampling methods is that $\Delta$ step changes for parameters with relational constraints are not guaranteed to be equivalent to those of other parameters.

We loosely addressed these problems by making a new aggregated parameter for each set of parameters that were related by constraints. We computed aggregated EEs for each trajectory by taking the mean absolute value of EEs for such parameters, resulting in a vector of $r$ EEs that was used in Equation 8 to compute $\hat{\mu}_p^*$ for each aggregated parameter. We considered these aggregated parameters as one parameter for the purpose of ranking parameter importance, and did not rank the original parameters.

### 2.5 Parameter Selection based on Bootstrapped Error

After the hydrological model runs completed for all trajectories, we estimated 90% confidence intervals for each parameter's $\hat{\mu}_p^*$ by bootstrapping. For each parameter, 1000 EE vectors of length $r$ had their elements sampled with replacement from the original $r$ EEs, and $\hat{\mu}_p^*$ was computed for each vector. We independently completed bootstrapping for each parameter (as in the SALib implementation by Herman and Usher, 2017) instead of sampling whole Morris trajectories (as in the STAR-VARS





implementation by Razavi and Gupta, 2016) to allow greater variation in the resulting quantile estimates, particularly for the analysis of aggregated parameters. We computed EEs for aggregated parameters by bootstrapping the original parameters' EE

values and aggregating them in the same manner as discussed in Section 2.4.1.

We used an EE cutoff to determine which parameters would be selected for calibration. First, for each sensitivity metric we determined the bootstrapped mean EE value corresponding to the top $X^{th}$ percentile, after removing parameters whose EEs were equal to zero and considering aggregated parameters as one. Then, we flagged all of the parameters whose estimated $95^{th}$ percentile EE values were greater than this cutoff value as being selected for calibration for that metric. We assume

all parameters within a selected aggregated parameter would be calibrated, but only report them as one parameter here. The union of parameters selected from all sensitivity metrics comprised the final set of calibration parameters. We evaluated the number of parameters selected as a function of the $X^{th}$ percentile cutoff for basin and hillslope outlet sensitivity analyses in Section 5. Subsequent results are presented for the $10^{th}$ percentile as an example cutoff; in practice the cutoff value should be defined separately for each sensitivity metric based on a meaningful change for the corresponding decision objective (e.g., the

$\epsilon$-tolerance in optimization problems (Laumanns et al., 2002)). To test the hypothesis of spatial variability in parameters that affect the sensitivity metrics, we compare parameters that would be selected based on each hillslope's EEs against each other and the basin outlet selection.

### 2.6 Evaluating the use of Parameter Multipliers

We compare the EEs for parameters that are traditionally adjusted by the same multiplier to determine if all parameter EEs

are meaningfully large and not statistically significantly different from each other. This would suggest a multiplier or another regularization method may be useful to reduce the dimensionality of the calibration problem. Otherwise, we recommend separately calibrating parameters whose EEs are large and statistically different from one another, as this suggests the multiplier would not uniformly influence the model outputs across adjusted parameters. We evaluate significance using the bootstrapped 90% confidence intervals.

### 235  3  Hydrologic Model Description: RHESSys

We used the Regional Hydro-Ecologic Simulation System (RHESSys) for this study (Tague and Band, 2004). RHESSys consists of coupled physically-based process models of the water, carbon, and nitrogen cycles within vegetation and soil storage volumes, and it completes spatially explicit water routing. Model outputs may be provided for patches (grid cells), hillslopes, and/or the basin outlet. We used a version of RHESSys adapted for humid, urban watersheds (Lin, 2019b), including water

routing for road storm drains and pipe networks, and anthropogenic sources of nitrogen. It also has modified forest ecosystem carbon and nitrogen cycles (a complete summary of modifications is provided in the README file). We used GIS2RHESSys (Lin, 2019a) to process spatial data into the modeling grid and file formats required to run RHESSys. GIS2RHESSys has several parameters that define how the RHESSys model is structured (e.g., locations of urban drainage, and grid cell resolution) but RHESSys model output sensitivity to these structural parameters is outside the scope of this paper. The full computational





workflow that was used for running GIS2RHESSys and RHESSys on the University of Virginia's Rivanna high performance
computer is provided in the code repository (Smith, 2021a).

For this paper, we classified RHESSys model parameters as structural or non-structural. A key structural modeling decision
is running the model in vegetation growth mode or in static mode, which only models seasonal vegetation cycles (e.g., leaf-on,
leaf-off), and net photosynthesis and evapotranspiration, and does not provide nitrogen cycle outputs. While authors Lin and
Band have developed a stable growth model for the study watershed, our analysis found that randomly sampling non-structural
growth model parameters within their specified ranges commonly resulted in unstable ecosystems (e.g., very large trees or
unrealistic mortality). It is beyond the scope of this paper to determine the conditions (parameter values) for which ecosystems
would be stable, so we used RHESSys in static mode. We used a statistical method to estimate total nitrogen (TN) as a function
of simulated streamflow, as described in Section 3.1. Other structural modeling decisions include using the Clapp-Hornberger
equations for soil hydraulics (Clapp and Hornberger, 1978), the Dickenson method of carbon allocation (Dickinson et al.,
1998), and the BiomeBGC leaf water potential curve (White et al., 2000). A full list is provided in a supplementary table (item
S2).

We categorized non-structural parameters according to the processes they control. Table 1 displays the parameter cate-
gories, processes, number of parameters in each category, and how many parameters can be adjusted by built-in multipliers.
A supplementary table (item S2) provides a full description of each parameter, the bounds of the uniform distribution used
for sensitivity analysis sampling, and justification for the parameter bounds. Hillslope and zone parameters control processes
over the entire modeling domain, while land use, vegetation, building, and soil parameters could be specified for each patch
modeled in RHESSys. Patch-specific parameter values for each category would result in more parameters than the number of
calibration data points, so we applied the same parameter values to each land use type (undeveloped, urban, septic), vegetation
type (grass and deciduous tree) and to buildings (exurban households), and grouped soil parameters by soil texture. To reduce
the number of parameters to calibrate, we did not consider specific tree species and their composition across the watershed
(e.g., Lin et al., 2019); all forest cover was modeled as broadleaf deciduous trees. Soil textures were classified as riparian or
non-riparian (referred to as "other" in this study). Because there is developed land, we further divided soil textures into uncom-
pacted or compacted for a total of four soil types (displayed in Fig. 3B). Given this coarse spatial resolution of the soils data,
we did not employ spatial sensitivity analysis methods that consider auto- and cross-correlations of soil parameter values (Koo
et al., 2020b; Lilburne and Tarantola, 2009).

RHESSys is typically calibrated using built-in parameter multipliers, which for this study would mean using 11 multipliers to
adjust 40 of the 271 possible parameters. While we know that some of these parameters are more easily measured than others,
we consider all 271 parameters in the sensitivity analysis. We aggregate parameters that are related by constraints, resulting
in 237 unique EEs for each sensitivity metric. Previous studies that implemented sensitivity analyses of RHESSys generally
adjusted a subset of the multipliers by limiting the analysis to process-specific parameters that are known or expected to affect
outputs of interest (e.g., streamflow in Kim et al. (2007), nitrogen export in Lin et al. (2015) and Chen et al. (2020), carbon
allocation in Garcia et al. (2016) and Reyes et al. (2017), and evapotranspiration and streamflow in Shields and Tague (2012)).





**Table 1.** Table of RHESSys parameter categories, the processes modeled in those categories for this study, the number of unique parameters in each category, and the number of parameters that can be adjusted by built-in RHESSys parameter multipliers.

| Parameter Category | Number of Parameters | Parameters Affected by Multipliers | Processes Controlled by Parameters |
|---|---|---|---|
| Hillslope | 2 | 2 | Controls how groundwater storage volumes are allocated to streams. |
| Land Use | 11 | 0 | Describes septic tank water loads, detention storage, and the imperviousness of each land cover type. |
| Soil | 104 | 36 | Defines soil property values that control hydraulic transport, and carbon and nitrogen cycles. |
| Vegetation | 135 | 2 | Defines vegetation property values that control radiation and moisture fluxes, and carbon and nitrogen cycles. |
| Buildings | 7 | 0 | Defined with vegetation parameters that control detention storage, height, and radiation fluxes. |
| Zone | 12 | 0 | Controls atmospheric processes across the watershed, including transmissivity, and temperature and precipitation lapse rates, which affect the assigned patch temperature and precipitation values across the watershed. |

Most of these studies used local one-at-a-time sensitivity analysis near a best estimate of parameter values from calibration or
prior information, with some exceptions that employed global sensitivity analyses (Lin et al., 2015; Reyes et al., 2017).

To our knowledge, this paper presents the first sensitivity analysis of all non-structural RHESSys model parameters. A global sensitivity analysis approach is used to discover which parameters and processes are most important to model streamflow for this study. Consequently, part of our discussion in Section 6 highlights those parameters that are selected for calibration based on the sensitivity analysis, yet are not adjusted using standard RHESSys multipliers or are otherwise uncommonly calibrated.
Even though the results are conditional on the specific parameter ranges (Shin et al., 2013), climatic input data and model outputs (Shields and Tague, 2012), and structural equations selected (Son et al., 2019), the resulting parameter identification should be generally useful to inform future studies that use RHESSys or other ecohydrologic models.

### 3.1 Modeling Total Nitrogen with WRTDS Regression

We used the Weighted Regression on Time Discharge and Season (WRTDS) method (Hirsch et al., 2010; Hirsch and De Cicco,
2015) to estimate daily total nitrogen (TN) concentration as a function of simulated streamflows. Equation 9 provides the





regression model

$$\ln(C_{TN,t}) = \beta_0 + \beta_1 \ln(Q_t) + \beta_2 t + \beta_3 \sin(2\pi t) + \beta_4 \cos(2\pi t) + \epsilon \tag{9}$$

where $C_{TN,t}$ is the TN concentration, $\beta_i$ is the $i^{th}$ regression model parameter, $Q_t$ is the streamflow (discharge), $t \in \mathbb{R}$ is a time index in years, and $\epsilon$ is residual error. The $\sin$ and $\cos$ terms model an annual cycle. We estimated regression model
parameters using the observed basin outlet streamflow and TN data. The parameter estimation procedure employs a local window approach to weight observations by their proximity in $t$, $Q_t$, and day of the year. Default values of these three WRTDS window parameters did not simulate the interquartile range of TN observations well, so we used a manual selection of WRTDS parameters to improve the model fit, as described in supplementary material (item S0). Furthermore, adding a quadratic log flow term did not result in a meaningful improvement, so we used the simpler Equation 9 model.

In order to use WRTDS for any streamflow value within the observation time period, we created two-dimensional $(t, Q_t)$ interpolation tables for each of the five model parameters and the residual error (provided in supplementary material item S6). Simulated flows that were outside of the observed range of values were assigned the parameters for the nearest flow value in the table. Zero flows were assigned zero concentration. These interpolation tables apply only to the concentration-streamflow relationship at the basin outlet. We did not estimate TN for hillslopes due to a concern that this basin outlet relationship would
overestimate TN in predominately forested hillslopes that would have different concentration-discharge relationships (Duncan et al., 2015) and in this watershed do not have septic tank sources of TN. As a result, parameter selection for hillslopes is limited to the three streamflow sensitivity metrics.

## 4    Case Study Site Description

We apply these methods to a RHESSys model of the Baisman Run watershed, which is an approximately 4 km² area that
is located about 20 km North-Northwest of Baltimore, Maryland, USA and is part of the larger Chesapeake Bay watershed (Fig. 3A inset map). Baisman Run was one of the Long Term Ecological Research sites for the Baltimore Ecosystem Study (Pickett et al., 2020), and has roughly 20 years of weekly water chemistry samples and daily streamflow samples measured at the watershed outlet. After a five year spin-up period, we completed sensitivity analysis for 2004-10-01 to 2010-09-30. The sensitivity analysis would screen parameters for calibration and validation using the additional years of data. There was
a drought and several large precipitation events in this time period that seemed representative of the remaining calibration dataset. The average annual precipitation total is about 1 m and the average monthly temperature ranges from -2 °C to 25 °C. The Baisman Run watershed is about 80% forested, and most trees are deciduous. Exurban development is located primarily in the headwaters where nearly all of the impervious surfaces are located (5% of the area). The remaining 15% of the watershed corresponds to grass vegetation, which is considered as a reforestation opportunity that could control flooding and reduce
nutrient exports. The goal of this sensitivity analysis is to inform the selection of parameters to calibrate a RHESSys model that could be used in such a reforestation optimization. We provide references to code and data used for this study as well as data processing notes in supplementary material (item S0).





## 5 Results

In Section 5.1 we present results for the six decision-relevant sensitivity metrics. In Section 5.1.1 we use these results to evaluate
the appropriateness of using multipliers for calibration. Finally, we compare results for calibration-relevant and decision-relevant metrics in Section 5.2.

### 5.1 Analysis for Decision-Relevant Sensitivity Metrics

We first evaluated selection of parameters for calibration based upon elementary effects (EEs) whose mean and $95^{th}$ percentile estimates were larger than the $X^{th}$ percentile of the set of all parameters' mean EEs. Figure 1 shows the total number of
unique parameters (out of 102 with non-zero EEs) that would be selected for calibration as a function of the top $X^{th}$ percentile cutoff value applied to the decision-relevant sensitivity metrics. The plotted total represents the union of the top $X$ percent across the six metrics for the basin outlet, and across the three streamflow metrics for hillslope outlets, so more than $X$ percent may be selected at each cutoff value. For hillslope outlets, the total is also computed over all hillslopes. The gap in number of parameters selected when using hillslope outlets instead of the basin outlet suggests that parameters that control physical
processes captured by the streamflow sensitivity metrics are different across the watershed, as explored further in Figure 3. For this problem, considering sensitivity metrics for hillslope outlets commonly results in an additional 10-20 parameters selected for calibration compared to only using the basin outlet. There can be as many as 40 more parameters near the $X = 50\%$ cutoff. For basin and hillslope outlets, the gap between using the bootstrapped $95^{th}$ percentile EE values instead of the mean values illustrates the importance of considering sampling uncertainty in parameter screening exercises. For this problem, sampling
uncertainty commonly adds 5-15 additional parameters. Near the $X = 50\%$ cutoff, almost all parameters would be selected for calibration using the hillslope outlets and $95^{th}$ percentile EE values. If desired, these sampling uncertainty gaps can be reduced by evaluating more Morris trajectories (e.g., by using progressive Latin hypercube sampling to add new trajectory starting points, as in Sheikholeslami and Razavi (2017)). This should bring the mean and $95^{th}$ percentile lines closer together in this figure.
For the selected 10% cutoff in Figure 1, 21 unique parameters would be selected for the basin outlet using the $95^{th}$ percentile EE values. Of these, 18 are selected based on the three streamflow metrics and 19 are selected based on the three TN metrics. This finding supports using sensitivity metrics for each output variable or objective of interest to select parameters to calibrate.

Basin outlet EEs are displayed in Figure 2 by parameter category (color) and type within each category (shape). Of the 237 parameters and aggregated parameters, 135 had EE values of exactly 0 for all metrics. These parameters primarily affect the
RHESSys nitrogen cycle and vegetation growth (which are not used in static mode), buildings, and some snow parameters. For streamflow sensitivity metrics (top row), differences in the selected parameters and their EEs across metrics suggest that flows of different magnitudes are affected by different physical processes, as expected (e.g., Ranatunga et al., 2016). For example, hillslope groundwater controls (index 1) and saturation to groundwater controls for compacted other soil (index 93) that affect how water moves from groundwater to riparian areas are selected parameters for each metric, but their EEs for low flows are
larger than for the other metrics. This is likely because groundwater would be the source of low flows. The EE magnitude for



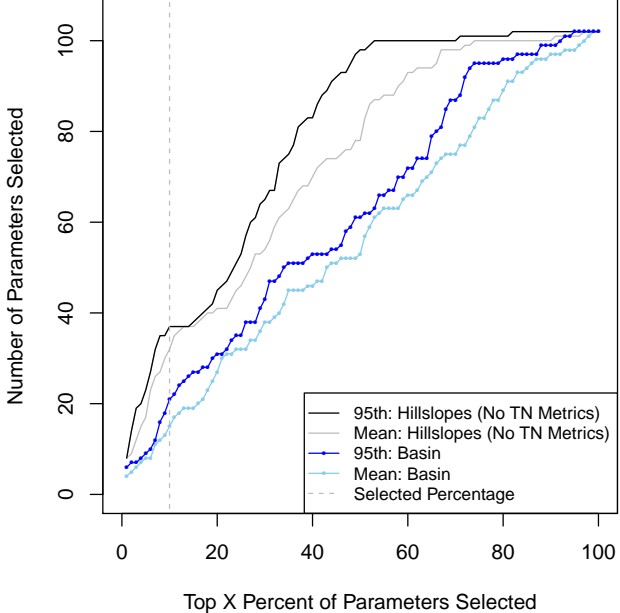

**Figure 1.** The number of parameters that would be selected for model calibration using the decision-relevant sensitivity metrics as a function of the cutoff percentage used to select parameters based on their elementary effects. The blue lines with circle points indicate the parameters that would be selected using only the basin outlet, while the gray lines correspond to using all hillslope outlets. Only streamflow metrics are considered for the hillslope outlets. Lighter line colors correspond to the mean and darker colors correspond to using the bootstrapped $95^{th}$ percentiles of the elementary effects to select parameters. The vertical dashed line indicates the selected 10% cutoff used as an example in this paper.

the specific rain capacity (interception storage capacity per leaf area index [LAI]) of trees (index 162) increases from flood flows to low flows. This result suggests that the impact of water intercepted by vegetation surfaces matters more for low flows, particularly in drought-stressed ecosystems, as that water alternatively reaching the ground would have a larger impact on the resulting stormflow hydrograph compared to a healthy ecosystem (e.g., Scaife and Band, 2017). Septic water loads (index 13),

which are modeled as constant throughout the year, have a higher mean EE for flood flows than the other streamflow metrics. This could result from uncertainty in saturated soil storage volumes leading to uncertainty in flood peaks. Similarly, the EE magnitude for tree maximum stomatal conductivity (index 119) is larger for flood flows, likely because of the impact on how quickly water can be transpired by trees. Finally, the EE for wind speed is largest for flood flows, which could be explained by the impact of wind on transpiration and the resulting reduction of the recessive limb of the hydrograph (e.g., Tashie et al.,

2019). Other parameters with larger EEs generally describe soil properties that are selected or are near the cutoff point for each streamflow metric. The largest of these for all metrics is the coefficient that describes bypass flow for other soils (index 73) which cover the largest area of the lower elevations in the watershed (Fig. 3B).

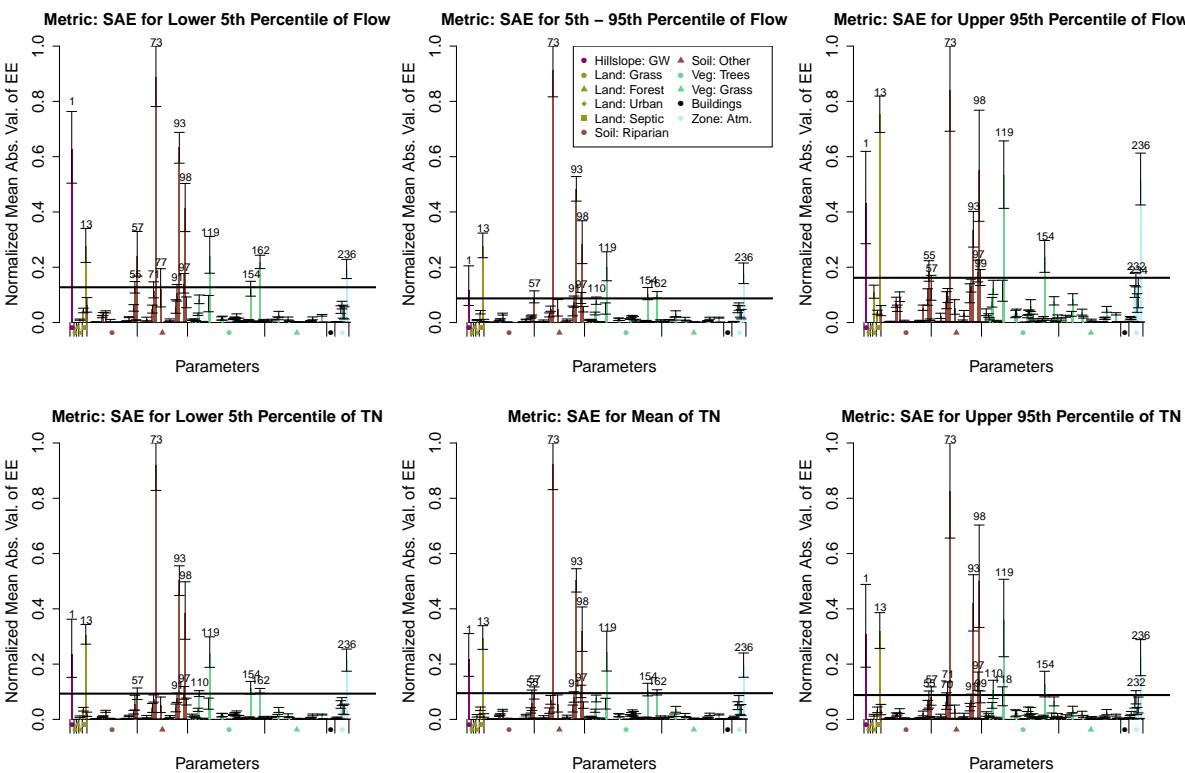

**Figure 2.** Mean absolute value of elementary effects for RHESSys model parameters evaluated for the six sensitivity metrics. Effects are normalized such that the maximum error bar value is equal to 1 on each plot. Colors indicate to which RHESSys category parameters belong, and symbols distinguish types within each category. Bootstrapped error bars extend from the $5^{th}$ to $95^{th}$ percentiles. Numbers above the error bars indicate the order along the x-axis for those parameters greater than the 10% cutoff (black horizontal line in each plot). Supplementary tables contain the data plotted in this figure (item S1).

For the three TN metrics (Fig. 2, bottom row), the parameters within the top 10 largest mean EEs are the same and their order is nearly identical when considering uncertainty. The largest EEs are close in magnitude to the $5^{th}$ to $95^{th}$ percentile

streamflow metric. These results make sense because the TN metrics are all affected by the same streamflows, and sample collection is often limited to low and moderate flow conditions (Shields et al., 2008). The only difference between the TN metrics is the selected WRTDS error quantile. EE error bars tend to be larger for the upper $95^{th}$ percentile TN estimate, which results in the selection of more parameters to calibrate. This result demonstrates the value of considering model error when selecting which parameters to calibrate.

For hillslope outlets, 37 unique parameters were selected using the 10% cutoff and the $95^{th}$ percentile EE values (Fig. 1). This parameter set contained all of the parameters identified using only the basin outlet. Those 37 parameters are listed in Figure 3C and 3D, which compare results for each hillslope and the basin outlet. Figure 3C provides the rank of mean EEs for





the upper $95^{th}$ percentile streamflow sensitivity metric. We provide plots for the other two streamflow sensitivity metrics in supplementary material (item S4). Figure 3D is aggregated over all decision-relevant sensitivity metrics (only streamflow for

hillslopes) and indicates whether or not the parameter would be selected for calibration. To guide the explanation of additional parameters selected from the hillslope analyses, Figure 3A and 3B respectively provide the land cover and soil texture types for the Baisman Run watershed. The majority of the watershed is forested. Impervious surfaces and grasses are primarily located in hillslopes 9 to 14 where exurban households are located. The two Southwest-Northeast trending linear features that appear as grass in Figure 3A and as compacted other soil S109 in Figure 3B correspond to power lines.

Figure 3C for the flood flow sensitivity metric shows that the previously described parameters with high mean EE ranks based on the basin outlet tend to also have high mean EE ranks in all hillslopes. Septic water load and riparian soil $m$ (hydraulic conductivity decay with saturation deficit) are exceptions, which only affect hillslopes with households and modeled riparian soils, respectively. Whether or not a hillslope is more forested or impervious explains many parameter rank differences among hillslopes (e.g., the percent impervious parameters). Tree parameters overall have higher ranks for more forested hillslopes, and

grass parameters have higher ranks in more impervious hillslopes, which also have more grass areas. Compacted soils S108 and S109 have higher parameter ranks in more impervious hillslopes where these soils have larger proportions of the total hillslope area relative to more forested hillslopes. Coverage area of riparian soils is less than other soils and these soils tend to be wet regardless of the conductivity value due to spatial position, which could explain why riparian parameters tend to have smaller ranks than other soil parameters. While it is not surprising that parameter EE ranks vary across the watershed according

to the hillslope features and respective processes that act in those areas (e.g., van Griensven et al., 2006; Herman et al., 2013b), this result demonstrates that evaluating sensitivity metrics across a watershed can lead to a different interpretation of which parameters are important to calibrate compared to evaluations completed for the outlet where calibration data are located.

Figure 3D further explores this point by showing which parameters would be selected for calibration using basin and hillslope analyses if aggregating the top 10% over all decision-relevant sensitivity metrics (only streamflow metrics for hillslopes).

Comparing the parameters selected in Figure 3D to their ranks for the flood flow sensitivity metric in Figure 3C reveals that some lower-ranked parameters for flood flow are ultimately selected for calibration. This result supports the use of multiple sensitivity metrics or objectives to select parameters. Furthermore, several parameters that would be selected for hillslope analyses would not be selected for the basin analysis if sensitivity metrics were not aggregated over space, with riparian soil parameters being the most common. Three tree parameters and both grass parameters were also selected for a few hillslopes

that are almost completely forested or have large grass areas, respectively, yet would not be selected for the basin analysis. Parameters that are selected for hillslopes but not for the basin would exert relatively smaller signals when calibrating to the basin outlet data, and would likely introduce equifinality to the calibration. However, there is value in considering such parametric uncertainty if the parameters have a meaningful contribution to the sensitivity of decision objectives nearer to the spatial scale of decision making (i.e., within the representative elementary watershed Reggiani et al., 1998). Specifically,

engineering designs that would affect flows at these spatial scales and locations ought to be robust to the parametric uncertainty in flows that would likely result from calibration of these parameters. This point is discussed further in Section 6.

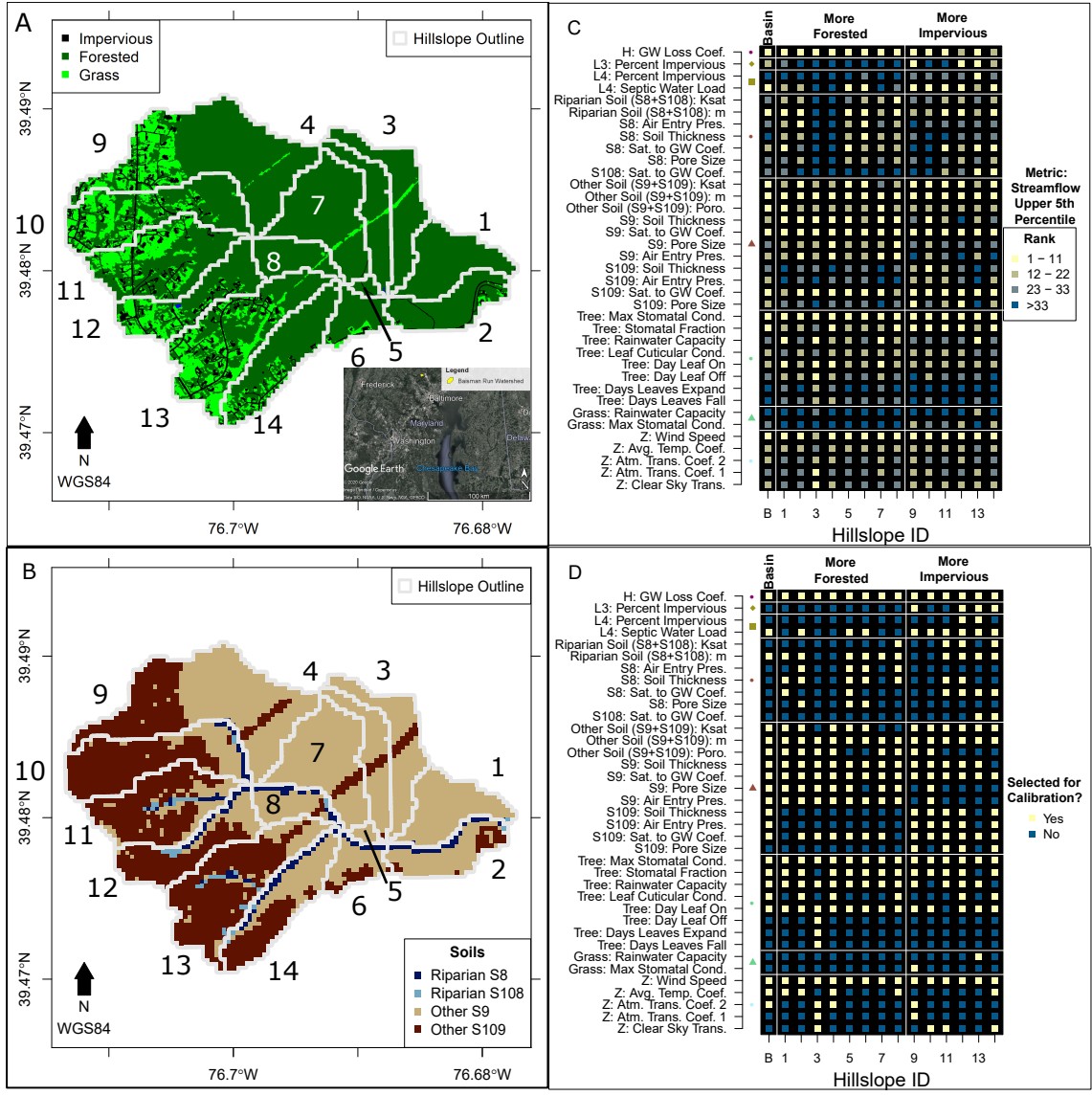

**Figure 3.** A: Land cover of the Baisman Run watershed (data provided by Chesapeake Conservancy, 2014), and an inset map showing the location in the U.S. (© Google Earth, 2020). Numbered hillslopes are outlined in gray. B: Soil types of Baisman Run (data provided by United States Department of Agriculture (USDA), 2017). Compacted soils begin with S10. C: Ranks of mean elementary effects for the $95^{th}$ percentile streamflow sensitivity metric for the basin outlet (B on x-axis) and each hillslope. Ranks are grouped by 11, which is 10% of the number of non-zero elementary effects. D: Indicators for whether or not a parameter would be selected for calibration, aggregated over all decision-relevant sensitivity metrics (only streamflow for hillslopes). In C and D, white horizontal lines divide parameter categories. Categories are labeled with symbols that match Figure 2. Vertical white lines divide the basin results from hillslope results, and more forested hillslopes from more impervious hillslopes. Abbreviations: GW - groundwater, Ksat - saturated hydraulic conductivity (cond.), m - describes cond. decay with sat., poro. - porosity, trans. - transmissivity.



### 5.1.1 Evaluation of Parameter Multipliers

We present results for only those multipliers whose adjusted parameters all have non-zero EEs. Figure 4 shows barplots of the bootstrapped mean and 90% confidence intervals of EEs for each of the ten multiplier parameters that could be used for the
selected RHESSys model structure. For EEs that were related by constraints ($m$ and hydraulic conductivity in Fig. 4) bars are plotted for their raw and aggregated values. These barplots correspond to the $95^{th}$ percentile streamflow sensitivity metric. We provide plots for the other five decision-relevant sensitivity metrics in supplementary material (item S5).

We evaluate the appropriateness of a parameter multiplier based on the magnitudes of the EEs and their uncertainty for parameters that can be adjusted by the same multiplier. Based on these considerations, a multiplier would not be recommended
for $m$ and the saturation to groundwater bypass flow coefficients (panels A and B), which show statistically significant differences in EE values across parameters in the set and at least one soil type with a large EE value. For specific leaf area (panel I), it would be preferable to simply calibrate the tree parameter instead of using a multiplier. For the maximum snow energy deficit (panel H), using one multiplier for riparian soils and another for other soils may be preferable. For all other parameters, a single multiplier or other regularization method could be used based on overlapping error bars and/or relatively small EE
values. These results hold well across the six decision-relevant sensitivity metrics and suggest that the dimensionality of the calibration could be reduced by employing parameter multipliers or another regularization method (e.g., Pokhrel and Gupta, 2010). Specifically for multipliers, if all 38 unaggregated parameters in this figure were selected for calibration, the aforementioned suggested multipliers could reduce the calibrated total to 15. Depending on the EE percentile cutoff used to select parameters (Fig. 1), the bottom row and possibly the middle row in Figure 4 may not be selected for calibration.

## 5.2 Analysis for Calibration-Relevant Sensitivity Metrics

Figure 5A provides plots of parameter EEs for the four calibration-relevant sensitivity metrics. The parameters with the largest EEs are nearly identical for the NSE, LNSE, and pBias metrics, and the EE magnitudes are closest to the $5^{th}$ to $95^{th}$ percentile streamflow metric (these metrics are highly correlated, as shown in supplementary item S7). Contrasting these results with Figure 2 suggests that the NSE and LNSE are not sufficient to capture parameters that affect flood and low flows, contrary
to reasoning often provided as justification for their use. The log-likelihood metric shows large EEs for many of the same parameters as other calibration and decision-relevant metrics; however, the magnitudes and rankings of parameters are different, and some new parameters are selected. Note that all parameters have non-zero EEs for the LogL metric as a result of equifinality in the parameters obtained from maximum likelihood estimation. The 10% threshold cutoff used to select parameters for calibration is larger than the resulting noise that is introduced into the EE values.
Figure 5B presents a plot indicating whether or not each parameter would be selected for calibration using the calibration-relevant and decision-relevant sensitivity metrics. Note that the calibration-relevant metrics did not identify any new parameters than the decision-relevant metrics evaluated across hillslopes (All, H), so the y-axis matches Figure 3C and 3D. Considering only basin outlet evaluations (All, B), decision-relevant metrics identify five parameters that the calibration-relevant metrics do not identify. These parameters include two atmospheric parameters that were selected from the flood flow decision metric,

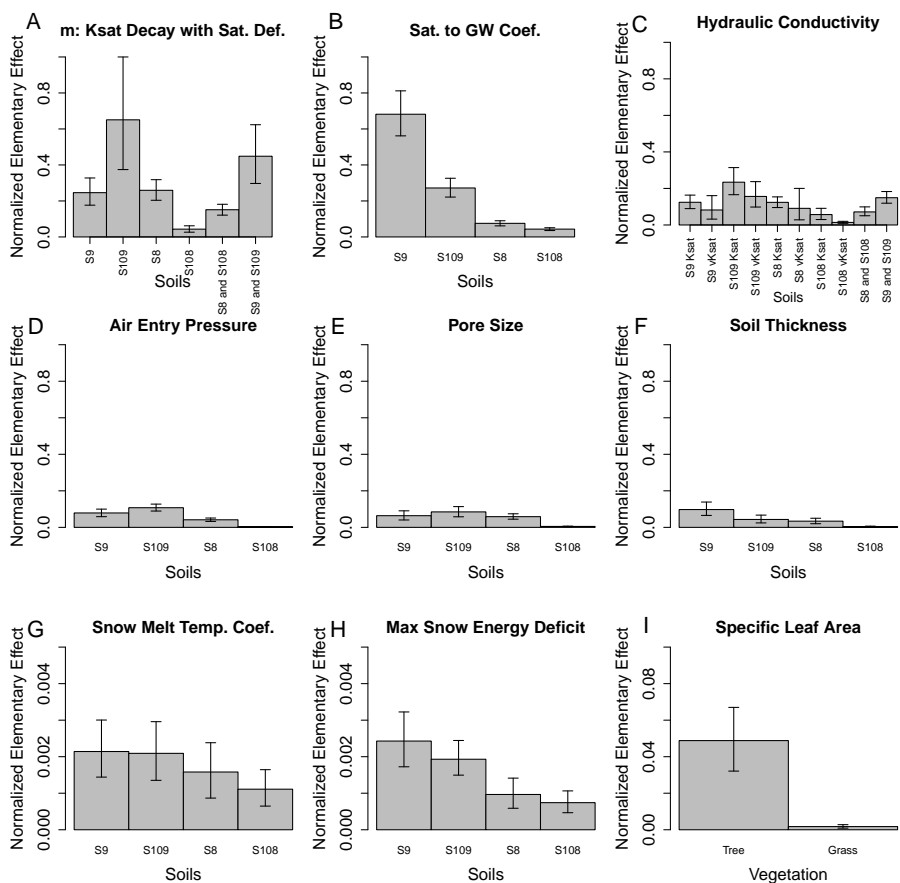

**Figure 4.** Barplots of the mean absolute value of the elementary effects for parameters that can be adjusted by ten RHESSys multiplier parameters (panel C contains two multipliers). Bootstrapped error bars extend from the $5^{th}$ to $95^{th}$ percentile estimates. The effects correspond to the $95^{th}$ percentile streamflow sensitivity metric, and are all normalized using the same maximum error bar value as in Figure 2. The x-axis of each plot indicates which soil or vegetation type is considered. For hydraulic conductivity, it also indicates which parameter is considered (vertical [vKsat] or lateral [Ksat] conductivity). Note that the plots in the bottom row have different y-axes ranges than each other and the plots above.

and a soil parameter that was selected from the low flow decision metric. The other two parameters were selected by considering model error in TN. Of the calibration-relevant metrics, only the log likelihood metric (LogL, B) identifies parameters that are unique from all other basin-evaluated metrics, but these parameters are selected for hillslopes using decision-relevant metrics (All, H). Of note is that a set of 10 parameters are selected for each of the calibration-relevant metrics and the aggregated decision-relevant metrics, and a set of 13 parameters are only selected from hillslope evaluation of the decision-relevant

streamflow metrics. This result strengthens the recommendation to spatially evaluate sensitivity metrics to inform parameter selection of spatially distributed models.





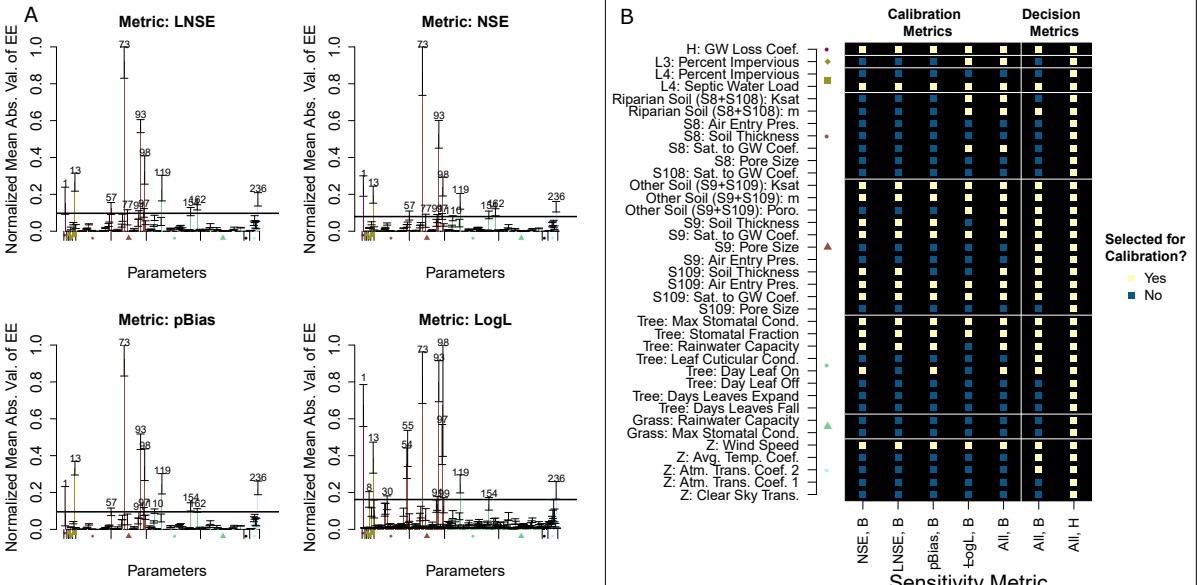

**Figure 5.** A: Mean absolute value of elementary effects for RHESSys model parameters evaluated for the four calibration-relevant sensitivity metrics. The style matches Figure 2. B: Indicators for whether or not a parameter would be selected for calibration for each of the calibration-relevant sensitivity metrics, and separately aggregated over all calibration-relevant and decision-relevant sensitivity metrics. B and H in the x-axis labels indicate basin outlet or hillslope outlet. Vertical white lines divide the calibration and decision-relevant sensitivity metric results. Other styles match Figure 3D.

# 6 Discussion

## 6.1 Importance of Decision-Relevant Sensitivity Metrics for Parameter Screening

When sensitivity analysis is used to inform model calibrations, a primary goal is usually to reduce the dimensionality of the
search space by screening those parameters that most affect the outputs to be calibrated. How model outputs are considered
in sensitivity analyses and subsequent screening exercises can affect which parameters are selected. We found that specifically
evaluating high and low flows as decision-relevant metrics revealed a different parameter selection than using the calibration-
relevant metrics that are often used to capture parameters that control such flows. While the NSE is mathematically sensitive
(i.e., not robust) to high flows, the EE magnitudes and parameters that are selected by the NSE sensitivity metric do not match
well with those selected from the high flows decision metric. Instead, the EE magnitudes and selected parameters resemble the
$5^{th}$ to $95^{th}$ percentile streamflow metric. A similar result is obtained for the LNSE metric. A possible explanation for these
results is that the high and low flows sensitivity metrics each represent only 5% of the timeseries used in the NSE and LNSE
metrics, while the $5^{th}$ to $95^{th}$ percentile metric represents 90% of the timeseries. This analysis demonstrates that parameter
selection based on decision objectives can result in different parameters than calibration objectives. Thus, these results support
future studies that would evaluate which parameter screening method is ultimately preferable for various decision problems.



This could be assessed by optimizing engineering designs for controlling high and low flows to models calibrated based on parameter screening informed by the two alternative approaches. By setting a synthetic true parameter set, one can evaluate whether or not there is a meaningful difference in performance of the solutions resulting from these two approaches compared to solutions designed to the synthetic true parameter set. However, calibration-relevant metrics have limited value for spatially-

distributed models because they can only be computed for gauged locations, so such studies would not be recommended for models of spatially heterogeneous regions without the supporting data. This highlights the value of more spatially distributed hydrologic monitoring to refine our understanding of which parameters control hydrologic processes throughout a watershed to inform engineering design.

To elaborate, sensitivity analyses that we completed for ungauged hillslope outlets led to the identification of more parame-

ters to calibrate than were selected based on sensitivity analysis at the gauged basin outlet. Calibrating additional parameters that have smaller impact at the gauged location is likely to exacerbate equifinality in simulated outputs. However, equifinality at the basin outlet will often result in variability in outputs at ungauged locations, such that calibration of these additional parameters should be important to better capture the physical processes in hillslopes where engineering controls could be located. Even if parameter values are unchanged from their prior distributions after calibration, locations of engineering control

measures can be optimized to be robust to the resulting uncertainty in model outputs across the watershed. Spatially distributed monitoring of model parameters and streamflow gauges within sub-catchments could help to reduce this uncertainty. In summary, spatial evaluation of sensitivity metrics for spatially distributed models allows for the discovery of parametric sources of uncertainty across the watershed to which engineering designs would have to be robust.

## 6.2 Determining Opportunities for Parameter Reduction

Spatial sensitivity analyses also reveal opportunities to reduce parametric uncertainty by using additional data and data types. Parametric uncertainty could be reduced for any parameter by better constraining its prior range. For example, septic water loads could be constrained with household water consumption surveys. Surveys and data collection efforts for other parameters can target those hillslopes for which model sensitivity is largest. Alternatively, some of the parameters that were identified as important for model calibration could instead be specified by additional input datasets to reduce the dimensionality of the

calibration search space. For example, impervious surface percentage could be specified spatially from the land cover dataset, and time series of wind speed may be obtained from weather gauges or satellite data and then be processed to the spatial scale of the model. These approaches would transfer parametric uncertainty to input data uncertainty, which would ideally be negligible. Finally, uncertainty may be reduced by better capturing spatial trends in parameter values. For example, using finer resolution soils data products, such as POLARIS estimates (Chaney et al., 2016), or implementing different vegetation species

composition in riparian and non-riparian areas. However, both of these approaches change the RHESSys model structure and add more parameters, so it is unclear if total uncertainty would be reduced, even if local hillslope performance is improved. Nevertheless, preliminary analysis with an uncalibrated RHESSys model in dynamic mode found that simulated streamflow and nitrogen were better aligned with observations when a more spatially explicit soil and vegetation parameterization was used (Lin (2021); vegetation by plant functional type is described in Lin et al. (2019)). Similar performance was observed for





soils data by Quinn et al. (2005) using RHESSys and by Anderson et al. (2006) using a SAC-SMA model. This lends support to future analyses that consider sensitivity analysis of alternative model structures and parameters to discover dominant processes, as in Mai et al. (2020) and Koo et al. (2020a). The selected parameters across water quantity and quality-focused metrics would likely be different if TN concentrations were estimated from a process-based model, as in the dynamic mode of RHESSys, instead of statistically as a function of streamflow using WRTDS (e.g., RHESSys and WRTDS estimations are compared in

Son et al. (2019)).

Parameter multipliers and other regularization methods are a common dimensionality reduction choice for spatially distributed models. A comparison of model sensitivity results for parameters that can be adjusted by built-in RHESSys multipliers revealed opportunities for dimensionality reduction, and also identified some parameters that may be better to calibrate individually for this problem. Local data collection could also help to reduce model sensitivity to these parameters. Future research

is needed to formally test these recommendations for their impact on model calibration.

For RHESSys streamflow simulations, the global sensitivity analysis identified some parameters for calibration that are not commonly calibrated and should therefore be assigned priors that are adjusted to local site conditions. For example, zone (atmospheric) parameters are typically assigned fixed site values, but this analysis suggests careful examination should be given to parameters that adjust the estimated average temperature based on the supplied minimum and maximum temperature time

series. For vegetation species simulated in static mode, this analysis revealed that stomatal and leaf conductivity parameters, interception storage capacity parameters, and the parameter that sets the first day leaves show on deciduous trees were among the most important for modeling streamflow. For primarily forested hillslopes, parameters describing the length of time that leaves open and fall are also important. In addition to these parameters that are not adjusted by built-in RHESSys multipliers, many of the soil and groundwater parameters that are adjusted by multipliers were also identified as important to calibrate, as

is common in practice.

### 6.3 Opportunities for Future Research

This paper focused on the importance of evaluating sensitivity analyses at the spatial scale and magnitude that is appropriate for decision making. Selecting the appropriate temporal resolution for the sensitivity metric and the time period of sensitivity analysis is also important to inform parameter selection. All of the sensitivity metrics in this paper are temporally aggregated

measures instead of time-varied. With this approach, two model runs could have very different simulated time series, yet could have similar metric values. Additionally, parameters that arise from different generating processes (e.g., floods from spring snowmelts vs. summer hurricanes) would not necessarily be parsed out from any one model run. For engineering problems, a magnitude-varying sensitivity analysis (Hadjimichael et al., 2020) could be useful to identify those parameters that control specific extremes in the objectives. A time-varying sensitivity analysis (Herman et al., 2013c; Meles et al., 2021) could discover

more seasonally important parameters. Related to this point, this sensitivity analysis was completed for a short 6-year period. For engineering designs that will last several decades, model sensitivity to alternative climate futures would be useful to identify additional parameters to calibrate that could become important in future climates, even if they are not historically important. Similar to the earlier discussion, considering uncertainty in these parameters for optimizations under future climatic conditions





would allow engineering designs to be robust to their uncertainty. Outside of an engineering context, Hundecha et al. (2020)

showed that selecting parameters that control processes within sub-catchments is important when using calibrated models for climate change forecasts.

A final consideration mentioned earlier for risk-based decision making is the use of deterministic or stochastic watershed models. Stochastic methods were not employed for this analysis due to the focus on comparing gauged and ungauged locations. However, sensitivity analysis for TN that considered quantiles of model residual error resulted in a different set of parameters

to calibrate relative to the mean. This result suggests that sensitivity analysis of stochastic watershed models could lead to different parameter selection based on the distribution of residual errors that would be required for stochastic engineering optimizations. Future work is needed to compare sensitivity analysis and resulting parameter selection for deterministic and stochastic watershed models.

## 7   Conclusions

This paper provided guidance on evaluating parametric model uncertainty at the spatial scales of interest for engineering decision-making problems. We used the results of a global sensitivity analysis to evaluate common methods to reduce the dimensionality of the calibration problem for spatially distributed hydrologic models. We found that the sensitivity of model outputs to parameters may be relatively large at ungauged sites where engineering control measures could be located, even though the corresponding sensitivity at the gauged location is relatively small. The spatial variation in parameters with the

largest sensitivity could be described well by variation in land cover and soil features, which suggests that different physical processes have important controls on model outputs across the watershed. If we select all parameters with the largest sensitivity metrics in ungauged locations for calibration, that will lead to more parameters compared to using only the gauged location. While the processes affected by the additional parameters would have a relatively small effect at the outlet location, thus exacerbating the equifinality problem during calibration, they would describe important variability in model outputs at the

engineering control locations. Thus, considering such parametric uncertainty in optimizations of engineering control measures should help to discover solutions that are robust to it. Sensitivity analysis results were also useful to inform which parameter multipliers may be useful to employ for further dimensionality reduction.

Results from this study support two critical avenues of future research that could further inform how to employ sensitivity analyses of models that are used in decision-making problems. The literature on sensitivity analysis of hydrologic models

almost exclusively corresponds to deterministic outputs, whereas a stochastic framework that considers model residual error should be and often is used to develop engineering designs. We found that considering model error resulted in selecting additional parameters to calibrate. Future research should formally compare sensitivity analysis of deterministic and stochastic watershed models that are employed for engineering decision making problems. Secondly, we found that the parameters screened by using common extreme streamflow calibration performance measures as sensitivity metrics do not match those pa-

rameters screened by specifically evaluating extreme flows. Future work should compare results of using screened parameters from each method to calibrate a model that is used in optimizing engineering controls to evaluate which method is ultimately





preferable for various decision problems, and whether or not there is a meaningful difference in performance of the resulting solutions.

*Code and data availability.* The exact code and data used for this study are made available in a HydroShare data repository (Smith, 2021b).
The code is tracked in the RHESSys_ParamSA-Cal-GIOpt GitHub repository (Smith, 2021a).

**Appendix A**

This appendix provides the probability density function (pdf) and the log-likelihood equations for the skew exponential power distribution that we used for the LogL sensitivity metric. We made minor changes to the equations presented in Schoups and Vrugt (2010) to apply their derivations to this problem, but most equations are identical. The pdf for a standardized skew
exponential power distributed variate, $a_t$, at time $t$ is described in Equation A1

$$f(a_t|\xi,\beta) = \frac{2\sigma_\xi\omega_\beta}{(\xi+\xi^{-1})}\exp^{-c_\beta|a_{\xi_t}|^{(\frac{2}{1+\beta})}} \tag{A1}$$

where $\xi$ is the skewness parameter and $\beta$ is the kurtosis parameter. Terms of the standard exponential power distribution are a function of $\beta$, as described in Equations A2 and A3

$$\omega_\beta = \frac{(\Gamma[\frac{3}{2}(1+\beta)])^{0.5}}{(1+\beta)(\Gamma[\frac{(1+\beta)}{2}])^{\frac{3}{2}}} \tag{A2}$$

$$c_\beta = (\frac{\Gamma[\frac{3}{2}(1+\beta)]}{\Gamma[\frac{1}{2}(1+\beta)]})^{(\frac{1}{1+\beta})}. \tag{A3}$$

Introducing skew into the standard exponential power distribution involves computing the mean and standard deviation of the skew-transformed variate, which are functions of the first (M1) and second (M2) absolute moments of the original distribution. These are described in Equations A4 to A7

$$\mu_\xi = M_1(\xi - \xi^{-1}) \tag{A4}$$

$$\sigma_\xi = -\sqrt{(M_2 - M_1^2)(\xi^2 + \xi^{-2}) + 2M_1^2 - M_2} \tag{A5}$$

$$M_1 = \frac{\Gamma[1+\beta]}{(\Gamma[\frac{3}{2}(1+\beta)]^{0.5})(\Gamma[\frac{1}{2}(1+\beta)])^{0.5}} \tag{A6}$$

$$M_2 = 1. \tag{A7}$$

The $a_{\xi_t}$ variable in A1 is defined in Equation A8

$$a_{\xi_t} = (\mu_\xi + \sigma_\xi a_t)\xi^{-\text{sign}(\mu_\xi + \sigma_\xi a_t)} \tag{A8}$$



where $a_t$ is defined from the streamflow residuals, $\epsilon_t$, that are computed after applying a magnitude-varying multiplier (Equation A9) that adjusted RHESSys simulated streamflows, as shown in Equation A10

$$\mu_t = \exp^{\mu_h |Q_t|} \tag{A9}$$

$$E_t = \mu_t Q_t \tag{A10}$$

where $Q_t$ is the simulated streamflow at time $t$ and $E_t$ is the adjusted streamflow. As a result of employing the multiplier, $\epsilon_t$ is

computed with respect to $E_t$. Our implementation modeled lag-1 autocorrelation, $\phi_1$, and heteroskedasticity (Equation A11) of $\epsilon_t$, which leads to $a_t$ being defined as in Equation A12

$$\sigma_t = \sigma_0 + \sigma_1 |E_t| \tag{A11}$$

$$a_t = \frac{\epsilon_t - \epsilon_{t-1} \phi_1}{\sigma_t} \tag{A12}$$

where $\sigma_t$ is the heteroskedasticity-adjusted standard deviation. From the above equations, there are six parameters that must be

estimated: $\beta$, $\xi$, $\sigma_0$, $\sigma_1$, $\phi_1$, and $\mu_h$. These are estimated by maximizing the log-likelihood provided in Equation A13

$$LogL = (T-1)\log\left(\frac{2\sigma_\xi \omega_\beta}{\xi + \xi^{-1}}\right) - c_\beta \sum_{t=2}^{T} |a_{\xi_t}|^{\left(\frac{2}{1+\beta}\right)} - \sum_{t=2}^{T} log(\sigma_t) \tag{A13}$$

where $T$ is the total number of data points in the time series. The first two terms result from Equation A1 and the final term results from the residual adjustment in Equation A12. Unlike the implementation in Schoups and Vrugt (2010), we begin at t = 2 so that no assumptions need to be made about the value of the t = 0 residual (which is both not simulated and unobserved).

We provide code that implements the maximum likelihood estimation in the code repository (the code is based on the spotpy Python package (Houska et al., 2015)), and provide fitting details in supplementary information (item S0).

*Author contributions.*  Jared D. Smith: Writing – original draft preparation, conceptualization, methodology, formal analysis, visualization, software, and data curation. Laurence Lin: Writing – review and editing, software, and data curation. Julianne D. Quinn: Writing – review and editing, conceptualization, methodology, visualization, and supervision. Lawrence E. Band: Writing – review and editing, conceptualization,

and supervision.

*Competing interests.*  The authors declare that they have no conflict of interest.

*Acknowledgements.*  The authors acknowledge Research Computing at The University of Virginia for providing computational resources and technical support that have contributed to the results reported within this publication. URL: https://rc.virginia.edu. The authors thank members of the Quinn and Band research groups for constructive feedback on this work. Data was supported by the Baltimore Ecosystem

Study.



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
