# Peer review of "Guidance on evaluating parametric model uncertainty at decision-relevant scales"

_Hydrology and Earth System Sciences, 2021_

## Author Comment (AC1)

**Authors' Response to Reviewers of "Guidance on evaluating parametric model uncertainty at decision-relevant scales" by Smith et al.**

Reviewer comments are in black

Author responses are in blue

Proposed text edits are in red

**RC1**

I have read with interest the manuscript entitled 'Guidance on evaluating parametric model uncertainty at decision-relevant scales'. The study examines the sensitivity of the simulations of a spatially distributed ecohydrological model to the model parameters for calibration purposes. Sensitivity is considered with respect to different model output metrics that correspond not only to performance metrics assessed at the basin outlet, but also statistics of the model output calculated for the different hillslopes of the basin where no output observations are available.

The study is a welcome contribution to the field of sensitivity analysis of spatially distributed models, which is challenging due to the high dimensionality of the parameter space of these models and which requires further investigations. The study discusses the issue of calibration and uncertainty estimation in absence of output observations, in particular at internal locations of a river basin where model-based information is critically needed to support decision making.

Overall, the manuscript is well written, the analyses were performed with care and the experiments are well documented in the supplements. However, I have a number of suggestions and I think that a number of points need clarification, in particular regarding the choice of (output) sensitivity metrics and the analysis of the multipliers, as detailed below.

Thank you for reviewing our manuscript and for your suggestions. We will adopt many of them into our revised version, as explained below.

p1 L8 'parameter multipliers': I suggest adding 'for spatially distributed parameters' for clarity.

We propose changing this sentence to:

We use global sensitivity analysis to screen parameters for model calibration, and to subsequently evaluate the appropriateness of using multipliers to adjust the values of spatially distributed parameters to further reduce dimensionality.

p1 L15-16 'for some parameter multipliers […] reducing dimensionality.': This needs clarification.

We agree this was confusing as written. Including the word "adjust" in the edit above should help to interpret this sentence. Additionally, we propose revising this sentence to directly reference the SA results, and also propose to add a second key point that we make about the use of multipliers.

3) for some multipliers, calibrating all parameters in the set being adjusted may be preferable to using the multiplier if they have significantly different parameter sensitivity values, while for others, calibrating only a subset of the parameters in the set may be preferable if they are not all influential.

p2 L31 'sensitivity metrics': I suggest specifying what this term refer to for clarity (e.g. performance measure or statistics o the simulated model output).

We propose removing metrics from this sentence. The next paragraph details what is meant by sensitivity metrics.

For studies that aim to use the resulting model to spatially optimize decisions, sensitivity should be defined for the decision objective values.

p2 L34-35: Could you provide some references/examples for this?

We propose citing studies from within the existing reference list as examples at the end of the sentence. Gupta and Razavi (2018) in section 1.2 describe different model performance metrics, and they cite many more studies as examples of each type of metric.

(e.g., Herman et al., 2013; van Griensven et al., 2006; Chen et al., 2020)

p2 L37-39 'Matching […] with controlling extremes.': A link between this sentence and the rest of the paragraph is missing.

We propose changing the previous sentence to:

Common calibration performance measures are used to quantify model performance across all flow magnitudes, yet some measures like the Nash-Sutcliffe Efficiency (NSE) lump several features together (Gupta et al., 2009) and specific features can govern the resulting performance value (e.g., peak flows for NSE in Clark et al., 2021).

The abuse of popular performance metrics in hydrologic modeling - Clark - - Water Resources Research - Wiley Online Library

Because our finding that the parameters selected by NSE more closely resemble $5^{th}$-$95^{th}$ percentile flow parameters than the $95^{th}$ percentile parameters, contrary to what might be expected by the results of the Clark et al. study, we propose also adding this sentence to the discussion:

For the Baisman Run watershed, parameters selected by the NSE may more closely resemble $5^{th}$ - $95^{th}$ percentile flow parameters because the flows >$95^{th}$ percentile are relatively small, and so the model residuals are a similar order of magnitude for peak flows and other flows.

p3 L92-93 'will be evaluated […] error model': The authors should clarify whether they refer here to future studies that may use the guidance presented in the manuscript.

Sorry for the confusion; we should have used the present tense as we evaluate sensitivity at ungauged locations in this paper. We propose re-writing this sentence:

We do not consider stochastic methods because we evaluate sensitivity in ungauged locations where no data are available to inform an error model.

p4 L107 'performance measure': Consider revising the terminology. The metric of Eq. (2) is not really a performance measure, as it does not use observed values.

That is correct that Eq. 2 is not actually error, so SAE is not an appropriate name for this. This should be sum of absolute median deviation (SAMD). We propose changing this terminology throughout the text. We also propose calling this a "relative variability measure" instead of a performance measure. For completeness, we propose computing the SAMD for the basin outlet as well, and comparing to the SAE in Eq 1.

p5 L130-131 'Therefore, […] the TN estimation method': This sentence is not clear to me. Observed data are also used for the streamflow objectives (Eq. (1)). In addition, does the water quality objective only consider the basin outlet or also hillslopes?

We do not use an error model for streamflow, but WRTDS as a regression model does have an error model (normal distribution for log[TN]). So, we could simulate the quantile estimates of TN in addition to the regression-predicted mean, and do sensitivity analysis on those quantiles. We propose modifying the existing text and added a clarifying sentence to explain this more clearly (starting in Line 127):

As described in Section 3.1, we used a linear regression model with normal residuals to estimate the log-space TN concentration at the outlet as a function of streamflow at the same location. As such, we could compute water quality sensitivity metrics for estimated quantiles from the regression error model, in addition to the regression-predicted mean. The water quality sensitivity metrics corresponded to 1) the 95th percentile of the distribution of estimated TN concentration, 2) the 5th percentile, and 3) the log-space mean (real-space median) on each of the days on which TN was sampled.

Yes, the water quality objectives are only evaluated at the basin outlet. This is solely because the WRTDS regression is valid only for the outlet. With a better spatial prediction model for TN, sensitivity could and should be evaluated across the catchment, as we do for streamflow.

p5 Section 2.3 A justification for the choice of the four calibration performance measures is missing (e.g. a justification could be that these metrics are typically used in previous studies, and in this case some references to some of these studies should be provided).

These are typical calibration metrics in hydrology. We propose changing the first sentence to:

Four performance measures that are typically used to calibrate hydrologic models are used...

And propose citing the following for NSE, pBias, and log-likelihood metrics:

Moriasi, D., Arnold, J., Van Liew, M., Bingner, R., Harmel, R., & Veith, T. (2007). Model evaluation guidelines for systematic quantification of accuracy in watershed simulations. Transactions of the ASABE, 50, 885.

Smith, T., Marshall, L., & Sharma, A. (2015). Modeling residual hydrologic errors with Bayesian inference. Journal of Hydrology, 528, 29-37. https://doi.org/10.1016/j.jhydrol.2015.05.051.

In addition, I think that the metric of Eq. (1) could also be used for calibration purposes. Therefore the difference between the decision-relevant and calibration-relevant sensitivity metrics is fuzzy.

Equation 1 is a performance measure, and that could be used for calibration. Decision relevance vs. calibration relevance is determined by which metrics are used (e.g., using only the lower 5th percentile of flows). We propose describing this more clearly by making section 2.2 "Decision-Relevant and Calibration-Relevant Sensitivity Metrics", and have sub-sub-sections for decision-relevant and calibration-relevant metrics (these are currently sub-sections 2.2 and 2.3). Some of the first paragraph of current sub-section 2.2 would be used in new sub-section 2.2, along with the following paragraph that defines performance measures, and decision and calibration relevant metrics:

Decision relevance is determined by sensitivity metrics that are evaluated on subsets of the timeseries that are relevant to decision objectives. While these metrics could be used for model calibration, that is an uncommon choice because the model would be unlikely to perform well on other data subsets (e.g., Efstratiadis and Koutsoyiannis, 2010). Calibration relevant metrics therefore use the full timeseries in this paper. Performance measures are a way to temporally aggregate a timeseries into a single value indicative of model fit to the observed data (e.g., Moriasi et al., 2007). The performance measures that we selected could all be used for model calibration, but the selected measure for decision-relevant metrics is not commonly used for calibration.

Efstratiadis, A. & Koutsoyiannis, D. (2010) One decade of multi-objective calibration approaches in hydrological modelling: a review, Hydrological Sciences Journal, 55:1, 58-78, DOI: 10.1080/02626660903526292

p6 L170 'EEs for each parameter […] in the parameter domain.': Please add a reference for this (e.g. Pianosi et al. (2016)).

We cited the original methods paper (Morris, 1991) in the paragraph above the equation.

p6 L178 'Step changes': Does this refer to the quantity Delta_{s+1,s,p} of Eq. (7)? Please clarify.

Yes, these are the deltas. We propose changing to:

Step changes, $\Delta$,

p6 L178-179 'to allow for a uniform […] within the specified bounds.': This needs to be better explained.

We do not know of a reference that explains this numerical method, so we propose adding a supplementary figure that illustrates this. We also propose changing the sentence to:

Step changes in parameter values were set to 50 levels, i.e. 50% of their range. For each parameter, this allows for a uniform distribution of parameter values across all samples (cite SI figure below).

[Figure]

Histograms for the Morris trajectory samples for parameter X as a function of the delta step size. Percentages are relative to the range of X.

p8 L217 'mean EE value': Clarify whether this refers to the metric defined in Eq. (8).

We propose changing to:

mean EE value (Eq. 8)

p8 Sect. 2.6: I do not understand this point. The use of a multiplier for a certain parameter type is based on the assumptions that the value of the parameters of this type vary in the same proportions in different locations of the basin, and I do not understand why these parameters should have similar sensitivity (here EEs).

If the parameters truly have a proportional adjustment, then it is still fine to use the multiplier even if the sensitivities are significantly different. We propose editing this section and parts of the results to say that parameters with significantly different sensitivity values *are candidates for* being calibrated individually. More investigation on the cause for their difference in sensitivity could inform the decision to calibrate individually or using a multiplier (e.g., is the difference in sensitivity caused by the parameters acting in vastly different proportions of the watershed area?)

p11 L300-303: This is based on a rather strong assumptions that the C-Q relationship obtained using the WRTDS regression method can be extrapolated to other flow conditions. This may not be appropriate for in-depth nitrogen study, in particular in agricultural areas with highly varying nitrogen inputs. I understand that it is not the point of this study to have a sophisticated nitrogen routine. However, I think that some comments should be added to further highlight the simplifications/limitations of the data-driven nitrogen routine used. My comment also refers to the sentence p21 L503-505.

In line 303, we propose modifying to:

Simulated flows that were outside of the observed range of values were assigned the parameters for the nearest flow value in the table. Extrapolation of the concentration-flow relationship to more extreme flows than were historically observed may provide inaccurate TN estimates, which is a limitation of this statistical prediction method. We expect the error from extrapolation in this basin to be low, as N loads appear to be dominated by effluent from septic systems as evidenced by isotopic sourcing (Kaushal et al., 2011, p. 8229), and septic effluent supply should be fairly steady over time.

Kaushal, S.S., Groffman, P.M., Band, L., Elliott, E.M., Shields, C.A., and Kendall, C. (2011). Tracking nonpoint source nitrogen pollution in human-impacted watersheds. Environmental Science & Technology, 19(45), pp. 8225-8232. https://doi.org/10.1021/es200779e

p13 L359 'healthy ecosystem': I suggesting replacing this expression by something like 'more humid ecosystem' for clarity.

We propose changing "compared to a healthy ecosystem" to:

compared to non-drought conditions

p14 L373-374: 'This result demonstrates […] to calibrate.': This sentence needs clarification.

We propose changing to:

The only reason for differences in which parameters are selected for calibration using the three TN metrics is uncertainty in the mean EE. EE error bars tend to be larger for the upper 95th percentile TN estimate, which results in the selection of more parameters to calibrate. This result demonstrates the value of considering both model error (different TN quantile estimates) and uncertainty in sensitivity (bootstrapped EE estimates) when selecting which parameters to calibrate. More parameters are found to be potentially influential when considering these sources of uncertainty.

p21 L511-514: The authors could refer to the study by Cuntz et al. (2016), which also demonstrates the importance of including in the calibration some parameters that are typically set to fixed values and in particular hard-coded parameters, using the NOAH-MP land surface model.

We agree this is a good paper to cite here.

p22 L553-555: I think that it should also be emphasized that, because of the issue equifinality, calibration strategies that identify an ensemble of possible parameter sets (as compared to a unique 'best' solution) and that therefore consider parameter uncertainty are more appropriate.

We propose modifying this sentence to:

Thus, due to equifinality, calibration methods that estimate parameter distributions are preferable to relying upon a single "best" parameter set; considering such parametric uncertainty in optimizations of engineering control measures should help to discover solutions that are robust to it.

Minor edits

p3 L61: I suggest replacing 'will' that 'are typically'.

Propose changing to "are used to inform"

p7 L192: replace 'smaller' by 'smallest'.

Propose accepting suggestion

References:

Cuntz, M., Mai, J., Samaniego, L., Clark, M., Wulfmeyer, V., Branch, O., et al. (2016). The impact of standard and hard-coded parameters on the hydrologic fluxes in the Noah-MP land surface model. Journal of Geophysical Research, 121(18), 10,676-10,700. https://doi.org/10.1002/2016JD025097

Pianosi, F., Beven, K., Freer, J., Hall, J. W., Rougier, J., Stephenson, D. B., & Wagener, T. (2016). Sensitivity analysis of environmental models: A systematic review with practical workflow. Environmental Modelling & Software, 79, 214–232. https://doi.org/10.1016/j.envsoft.2016.02.008

---

## Author Response (AR1)

**Authors' Response to Editor and Reviewers of "Guidance on evaluating parametric model uncertainty at decision-relevant scales" by Smith et al.**

Editor and Reviewer comments are in black. Note that many of the actual changed text edits are the same as proposed in our replies to reviewer comments. We have noted new edits with {highlighted text} preceding the reviewer comment. Line numbers in our replies correspond to the revised manuscript PDF without tracked changes in it.

Author responses are in blue

Text edits are in red

**Editor:**

Dear authors,
The manuscript has now received two reviews. Both reviewers see merit in this manuscript, though both have raised several points needing clarification.

I've thoroughly read the responses and proposed revisions from the authors, and find them to be a thoughtful treatment of planned changes. I appreciate that the authors separated their responses and their proposed text edits.

A majority of the comments from the reviewers, especially reviewer 2, encourage being explicit around the framing of the manuscript and the major outcomes from the manuscript. I encourage the authors to place particular effort to clarify these points of confusion raised by the reviewers. Overall, the manuscript is well written, and minor clarifications encouraged by the reviewers will improve its impact.

I do think that the major points outlined by the authors in their abstract and throughout the manuscript make a notable contribution to the literature, particularly the literature connecting sensitivity analysis and decision-making. However, I agree with the reviewers that it is worth investing time to ensure that these key messages are clear, and can be interpreted through your figures. As pointed out by one reviewer, it can be challenging to interpret the figures. One option may be to add a visual guide within each figure on how to read/interpret some of your figures (e.g., Fig 4, Fig 5), to reduce the amount of information shown (e.g., reduce the numbers included in Figure 5 to highlight key information), and/or to simply show larger figures or break subplots up (e.g., Figure 2, Figure 5). I really want to dissect Figure 2 and Figure 5, but am struggling to extract information from these given there is so much information packed in, as noted by one of the reviewers.

I encourage the authors to proceed with a thorough revision and I look forward to reading an updated manuscript.

Thank you for your review of our manuscript. We have implemented the proposed changes in response to reviewers and documented them in this response. There are other minor text edits as tracked changes in the PDF.  Additionally, we adopted several figure changes to improve their presentation:

Figure 2: Font size for the numbers and legend labels have been increased by 50%, and the figure is now on a full page. The legend has been moved outside of the plot area. A table has been added for the parameters corresponding to the numbers above the error bars.

Figure 4: y axis labels were removed for all but one sub-plot. Increased image size.

Figure 5: The original Figure 5A and 5B were split into Figure 5 and 6. Both figures were enlarged relative to 5A and 5B. The new Figure 5 x and y axes' labels were improved.

**RC1**

I have read with interest the manuscript entitled 'Guidance on evaluating parametric model uncertainty at decision-relevant scales'. The study examines the sensitivity of the simulations of a spatially distributed ecohydrological model to the model parameters for calibration purposes. Sensitivity is considered with respect to different model output metrics that correspond not only to performance metrics assessed at the basin outlet, but also statistics of the model output calculated for the different hillslopes of the basin where no output observations are available.

The study is a welcome contribution to the field of sensitivity analysis of spatially distributed models, which is challenging due to the high dimensionality of the parameter space of these models and which requires further investigations. The study discusses the issue of calibration and uncertainty estimation in absence of output observations, in particular at internal locations of a river basin where model-based information is critically needed to support decision making.

Overall, the manuscript is well written, the analyses were performed with care and the experiments are well documented in the supplements. However, I have a number of suggestions and I think that a number of points need clarification, in particular regarding the choice of (output) sensitivity metrics and the analysis of the multipliers, as detailed below.

Thank you for reviewing our manuscript and for your suggestions. We adopted many of them into our revised version, as explained below.

p1 L8 'parameter multipliers': I suggest adding 'for spatially distributed parameters' for clarity.

We changed this sentence to:

L7-9:

We use global sensitivity analysis to screen parameters for model calibration, and to subsequently evaluate the appropriateness of using multipliers to adjust the values of spatially distributed parameters to further reduce dimensionality.

p1 L15-16 'for some parameter multipliers […] reducing dimensionality.': This needs clarification.

We agree this was confusing as written. Including the word "adjust" in the edit above should help to interpret this sentence. Additionally, we revised this sentence to directly reference the SA results, and also added a second key point that we make about the use of multipliers.

L15-17:

3) for some multipliers, calibrating all parameters in the set being adjusted may be preferable to using the multiplier if they have significantly different parameter sensitivity values, while for others, calibrating only a subset of the parameters in the set may be preferable if they are not all influential.

p2 L31 'sensitivity metrics': I suggest specifying what this term refer to for clarity (e.g. performance measure or statistics o the simulated model output).

We removed "metrics" from this sentence. The next paragraph details what is meant by sensitivity metrics.

L32-33:

For studies that aim to use the resulting model to spatially optimize decisions, sensitivity should be defined for the decision objective values.

p2 L34-35: Could you provide some references/examples for this?

We now cite studies from within the existing reference list as examples at the end of the sentence. Gupta and Razavi (2018) in section 1.2 describe different model performance metrics, and they cite many more studies as examples of each type of metric.

L39-40:

(e.g., Herman et al., 2013; van Griensven et al., 2006; Chen et al., 2020)

p2 L37-39 'Matching […] with controlling extremes.': A link between this sentence and the rest of the paragraph is missing.

We changed the previous sentence to:

L40-44:

Common calibration performance measures are used to quantify model performance across all flow magnitudes, yet some measures like the Nash-Sutcliffe Efficiency (NSE) lump several features of the hydrologic time series together (Gupta et al., 2009) and specific features can govern the resulting performance value (e.g., peak flows for NSE in Clark et al., 2021).

The abuse of popular performance metrics in hydrologic modeling - Clark - - Water Resources Research - Wiley Online Library

Because our finding that the parameters selected by NSE more closely resemble $5^{th}$-$95^{th}$ percentile flow parameters than the $95^{th}$ percentile parameters, contrary to what might be expected by the results of the Clark et al. study, we also added this sentence to the discussion:

L490-492:

Another possibility is that in the Baisman Run watershed, flows greater than the $95^{th}$ percentile are still relatively small, and so the model residuals are a similar order of magnitude for peak flows and other flows.

p3 L92-93 'will be evaluated […] error model': The authors should clarify whether they refer here to future studies that may use the guidance presented in the manuscript.

Sorry for the confusion; we should have used the present tense as we evaluate sensitivity at ungauged locations in this paper. We propose modified this sentence to:

L97-98:

We do not consider stochastic methods because we evaluate sensitivity in ungauged locations where no data are available to inform an error model.

{new analysis, paragraph, and supplementary figure} p4 L107 'performance measure': Consider revising the terminology. The metric of Eq. (2) is not really a performance measure, as it does not use observed values.

That is correct that Eq. 2 is not actually error, so SAE is not an appropriate name for this. This should be sum of absolute median deviation (SAMD). We changed this terminology throughout

the text. We also called this a "relative variability measure" instead of a performance measure. For completeness, we computed the SAMD for the basin outlet as well, and compared results to the SAE (supplementary figure below in low res).

L117-124:

For the basin outlet, we used the sum of absolute error (SAE) as the performance measure for decision-relevant sensitivity metrics. Because performance measures require an observation time series to compute, we needed a different approach to measure relative variability for hillslope sensitivity analysis. We used the sum of absolute median deviation (SAMD), where the median value was computed across all model simulations of each hillslope. For completeness, we also used the SAMD for the basin outlet and compared to the SAE results in supplementary material (item S9). We found similar parameter selection and sensitivity ranking results for each method, which demonstrates that an observation time series is not necessary to obtain the parameter set to calibrate, although observations help to check that SA model simulations are reasonable. In this paper, we present basin outlet results for the SAE. The SAE and SAMD expressions are shown in Equations 1 and 2.

[Figure]

p5 L130-131 'Therefore, […] the TN estimation method': This sentence is not clear to me. Observed data are also used for the streamflow objectives (Eq. (1)). In addition, does the water quality objective only consider the basin outlet or also hillslopes?

We do not use an error model for streamflow, but WRTDS as a regression model does have an error model (normal distribution for log[TN]). So, we could simulate the quantile estimates of TN in addition to the regression-predicted mean, and do sensitivity analysis on those quantiles. We modified the existing text and added a clarifying sentence to explain this more clearly:

L141-147:

As described in Section 3.1, we used a linear regression model with normal residuals to estimate the log-space TN concentration at the outlet as a function of time, season, and streamflow at the same location. As such, we could compute water quality sensitivity metrics for estimated quantiles from the regression error model, in addition to the regression-predicted mean. The water quality sensitivity metrics corresponded to 1) the 95th percentile of the distribution of estimated TN concentration, 2) the 5th percentile, and 3) the log-space mean (real-space median) on each of the days on which TN was sampled.

Yes, the water quality objectives are only evaluated at the basin outlet. This is solely because the WRTDS regression is valid only for the outlet. With a better spatial prediction model for TN, sensitivity could and should be evaluated across the catchment, as we do for streamflow.

p5 Section 2.3 A justification for the choice of the four calibration performance measures is missing (e.g. a justification could be that these metrics are typically used in previous studies, and in this case some references to some of these studies should be provided).

These are typical calibration metrics in hydrology. Later in the paragraph, we state that these metrics are used to represent different features of the hydrologic time series. We changed the first sentence to:

L151:

Four performance measures that are typically used to calibrate hydrologic models are used...

And added the following citations for NSE, pBias, and log-likelihood metrics:

Moriasi, D., Arnold, J., Van Liew, M., Bingner, R., Harmel, R., & Veith, T. (2007). Model evaluation guidelines for systematic quantification of accuracy in watershed simulations. Transactions of the ASABE, 50, 885.

Smith, T., Marshall, L., & Sharma, A. (2015). Modeling residual hydrologic errors with Bayesian inference. Journal of Hydrology, 528, 29-37. https://doi.org/10.1016/j.jhydrol.2015.05.051.

{==new paragraph==}In addition, I think that the metric of Eq. (1) could also be used for calibration purposes. Therefore the difference between the decision-relevant and calibration-relevant sensitivity metrics is fuzzy.

Equation 1 is a performance measure, and that could be used for calibration. Decision relevance vs. calibration relevance is determined by which metrics are used (e.g., using only the lower $5^{th}$ percentile of flows). We have described this more clearly by making section 2.2 "Sensitivity Metrics" with two sub-sections for decision-relevant (2.2.1) and calibration-relevant (2.2.2) metrics (these were formerly sections 2.2 and 2.3). The following paragraph composes Section 2.2, which defines performance measures and decision- and calibration-relevant metrics:

L105-115:

In many hydrological studies, sensitivity analysis is used to understand how input parameters influence model performance measures (Jackson et al., 2019), such as the Nash-Sutcliffe efficiency. Performance measures are a way to temporally aggregatea time series into a single value that is indicative of model fit to the observed data (e.g., Moriasi et al., 2007). Gupta and Razavi(2018) note that using such performance measures as sensitivity metrics amounts to a parameter identification study to discoverwhich parameters may be adjusted to improve model fit. Therefore, the calibration-relevant sensitivity metrics in this paper usesuch performance measures on the full time series. Evaluating performance measures for subsets of the time series that describe specific features of interest (Olden and Poff, 2003) should identify those parameters that control processes that generate thosefeatures (e.g., timing vs. volume metrics in Wagener et al., 2009). Therefore, decision-relevant sensitivity metrics are evaluatedon subsets of the time series that are relevant to decision objectives. While these metrics could be used for model calibration,that is an uncommon choice because the model would be unlikely to perform well on other data subsets (e.g., Efstratiadis andKoutsoyiannis, 2010). The following subsections present the decision- and calibration-relevant sensitivity metrics.

Efstratiadis, A. & Koutsoyiannis, D. (2010) One decade of multi-objective calibration approaches in hydrological modelling: a review, Hydrological Sciences Journal, 55:1, 58-78, DOI: 10.1080/02626660903526292

p6 L170 'EEs for each parameter [...] in the parameter domain.': Please add a reference for this (e.g. Pianosi et al. (2016)).

We cited the original methods paper (Morris, 1991) in the paragraph above the equation.

p6 L178 'Step changes': Does this refer to the quantity Delta_{s+1,s,p} of Eq. (7)? Please clarify.

Yes, these are the deltas. We changed this to:

L196:

Step changes, Δ,

p6 L178-179 'to allow for a uniform […] within the specified bounds.': This needs to be better explained.

We do not know of a reference that explains this numerical method, so we added a supplementary figure that illustrates this. We also changed the sentence to:

L196-198:

Step changes, Δ, in parameter values were set to 50 levels (i.e. 50% of their range). For each parameter, this allows for a uniform distribution of parameter values across all samples (example sampling distributions for other percentages are provided in supplementary item S8).

[Figure]

Histograms for the Morris trajectory samples for parameter X as a function of the delta step size. Percentages are relative to the range of X.

p8 L217 'mean EE value': Clarify whether this refers to the metric defined in Eq. (8).

We changed this to:

L236:

mean EE value (Eq. 8)

p8 Sect. 2.6: I do not understand this point. The use of a multiplier for a certain parameter type is based on the assumptions that the value of the parameters of this type vary in the same proportions in different locations of the basin, and I do not understand why these parameters should have similar sensitivity (here EEs).

Now Section 2.5

If the parameters truly have a proportional adjustment, then it is still fine to use the multiplier even if the sensitivities are significantly different. We edited this section and parts of the results to say that parameters with significantly different sensitivity values *are candidates for* being calibrated individually. More investigation on the cause for their difference in sensitivity could inform the decision to calibrate individually or using a multiplier (e.g., is the difference in sensitivity caused by the parameters acting in vastly different proportions of the watershed area?)

p11 L300-303: This is based on a rather strong assumptions that the C-Q relationship obtained using the WRTDS regression method can be extrapolated to other flow conditions. This may not be appropriate for in-depth nitrogen study, in particular in agricultural areas with highly varying nitrogen inputs. I understand that it is not the point of this study to have a sophisticated nitrogen routine. However, I think that some comments should be added to further highlight the simplifications/limitations of the data-driven nitrogen routine used. My comment also refers to the sentence p21 L503-505.

Note that our study watershed is not agricultural and N loads appear to be dominated by septic systems. We edited this to:

L322-327:

Simulated flows that were outside of the observed range of values were assigned the parameters for the nearest flow value in the table. Extrapolation of the concentration-flow relationship to more extreme flows than were historically observed may provide inaccurate TN estimates, which is a limitation of this statistical prediction method. We expect the error from extrapolation in this basin to be low, as N loads appear to be dominated by effluent from septic systems as evidenced by isotopic sourcing (Kaushal et al., 2011, p. 8229), and septic effluent supply should be fairly steady over time.

Kaushal, S.S., Groffman, P.M., Band, L., Elliott, E.M., Shields, C.A., and Kendall, C. (2011). Tracking nonpoint source nitrogen pollution in human-impacted watersheds. Environmental Science & Technology, 19(45), pp. 8225-8232. https://doi.org/10.1021/es200779e

p13 L359 'healthy ecosystem': I suggesting replacing this expression by something like 'more humid ecosystem' for clarity.

We changed "compared to a healthy ecosystem" to:

L383:

compared to non-drought conditions

p14 L373-374: 'This result demonstrates […] to calibrate.': This sentence needs clarification.

We changed this to:

L396-401:

The reason for differences in which parameters are selected for calibration using the three TN metrics is uncertainty in the mean EE. EE error bars tend to be larger for the upper 95th percentile TN estimate, which results in the selection of more parameters to calibrate. This result demonstrates the value of considering both model error (different TN quantile estimates) and uncertainty in sensitivity (bootstrapped EE estimates) when selecting which parameters to calibrate. More parameters are found to be potentially influential when considering these sources of uncertainty.

p21 L511-514: The authors could refer to the study by Cuntz et al. (2016), which also demonstrates the importance of including in the calibration some parameters that are typically set to fixed values and in particular hard-coded parameters, using the NOAH-MP land surface model.

We have cited this paper here.

p22 L553-555: I think that it should also be emphasized that, because of the issue equifinality, calibration strategies that identify an ensemble of possible parameter sets (as compared to a unique 'best' solution) and that therefore consider parameter uncertainty are more appropriate.

We modified this sentence to:

L585-587:

Thus, due to equifinality, calibration methods that estimate parameter distributions are preferable to relying upon a single "best" parameter set; considering such parametric uncertainty in optimizations of engineering control measures should help to discover solutions that are robust to it.

Minor edits

p3 L61: I suggest replacing 'will' that 'are typically'.

Changed to "are used to inform"

p7 L192: replace 'smaller' by 'smallest'.

Accepted suggestion

References:

Cuntz, M., Mai, J., Samaniego, L., Clark, M., Wulfmeyer, V., Branch, O., et al. (2016). The impact of standard and hard-coded parameters on the hydrologic fluxes in the Noah-MP land surface model. Journal of Geophysical Research, 121(18), 10,676-10,700. https://doi.org/10.1002/2016JD025097

Pianosi, F., Beven, K., Freer, J., Hall, J. W., Rougier, J., Stephenson, D. B., & Wagener, T. (2016). Sensitivity analysis of environmental models: A systematic review with practical workflow. Environmental Modelling & Software, 79, 214–232. https://doi.org/10.1016/j.envsoft.2016.02.008

**RC2**

1. L8-10: "We evaluate six sensitivity metrics that align with four decision objectives; two metrics consider model residual error that would be considered in spatial optimizations of engineering designs." This sentence is confusing -- do the six sensitivity metrics add up to the four decision objectives + two model residual error metrics?

   Yes, that is correct. We modified the sentence to:

   L9-10:

   We evaluate six sensitivity metrics, four of which align with decision objectives and two of which consider model residual error…

2. L39: extremes or high flows?

Engineering controls can work to lower high flows and raise low flows. We changed this to:

L45:

Extreme high and low flows

3. It would be helpful to somewhere define "engineering controls".

   We added examples to the first sentence in the introduction.

   (e.g., green and gray infrastructure)

4. L 116: here are you write flooding, low flow, reservoir water supply objectives, but earlier you wrote flooding, low flow, and all other flows. If these are the same, that should be explicitly stated.

   We changed L131 to:

   flooding, low flow, and all other flow objectives

5. L320: "The goal of this sensitivity analysis is to inform the selection of parameters to calibrate a RHESSys model that could be used in such a reforestation optimization." is this the overall goal of this paper? If so, this should be stated in the introduction section much earlier.

   We added the following sentence at the end of the first paragraph of the introduction:

   L35-37:

   In this paper, we evaluate the influence of decision-relevant and calibration-relevant sensitivity metrics on parameter selection for calibration, and discuss the potential implications on subsequent model calibration and optimization of water management decisions.

6. L 328: are the elementary effects for all the parameters normalized on a percentage basis? Why compare the 95th percentile for the elementary effects to the overall mean of all

parameters' elementary effects, if that is what is being explained in this sentence? What does the 95th percentile estimate for the elementary effects mean?

As explained in Section 2.4, we completed bootstrapping of the elementary effects to generate a distribution of mean absolute values of elementary effects for each parameter. This recognizes that there is uncertainty in our estimate of a parameter's mean absolute elementary effect. From that distribution, we obtain the 95th percentile estimate of the mean absolute value EE for each parameter. Normalization in each panel of figure 2 is completed by taking the maximum 95th percentile EE value across all parameters and setting that to 1 (i.e., all EEs are divided by this value). The minimum is 0. So, the figure normalization is not on a percentage basis. It should also be noted that the EE is a normalized metric to begin with, as it is the absolute value of the change in the output metric per change in the input parameter, where the change in the input parameter is 50% of its range in our study.

Also stated in Section 2.4, we do not compare 95th percentile EEs to the overall mean across all parameters EEs. We sort the mean absolute EE values across parameters from largest to smallest and find the top X%. We then compare each parameter's 95th percentile estimate of its mean absolute EE to the X%-ile. Any parameter whose 95th percentile mean absolute EE estimate is above that threshold is selected for calibration.

7. Figure 3 seems to be referred to before figure 2 (L335).

   This reference to Figure 3 is to state that we will discuss a point in more detail later in the same section. We plan to leave this as-is unless asked to change.

8. L349: does an elementary effect value of exactly 0 mean that this parameter has no effect on the stream flow or hillslope metric? It would be helpful to state this explicitly.

   Yes, it does. We added the following to the end of the sentence:

   L373-374:

   (i.e., these parameters do not affect model-predicted streamflow)

9. {new table} The text discussing figure 2 is useful, (line 348 in the rest of this paragraph), but without knowing what the specific parameter numbers are in figure 2, I'm not sure what to take from this graphic.

The supplementary material provides the full list in number order in a spreadsheet. We discuss the parameters with the largest elementary effects within the text that you mention.

We have also included a table of the numbered parameters within Figure 2.

| Number | Parameter Name |
|--------|----------------|
| 1 | H: GW Loss Coef. |
| 13 | L4: Septic Water Load |
| 55 | Riparian Soil (S8+S108): m |
| 57 | S9: Soil Thickness |
| 70 | S9: Pore Size |
| 71 | S9: Air Entry Pres. |
| 73 | S9: Sat. to GW Coef. |
| 77 | S109: Soil Thickness |
| 91 | S109: Air Entry Pres. |
| 93 | S109: Sat. to GW Coef. |
| 97 | Other Soil (S9+S109): Ksat |
| 98 | Other Soil (S9+S109): m |
| 99 | Other Soil (S9+S109): Poro. |
| 110 | Tree: Day Leaf On |
| 118 | Tree: Leaf Cuticular Cond. |
| 119 | Tree: Max Stomatal Cond. |
| 154 | Tree: Stomatal Fraction |
| 162 | Tree: Rainwater Capacity |
| 232 | Z: Avg. Temp. Coef. |
| 234 | Z: Atm. Trans. Coef. 2 |
| 236 | Z: Wind Speed |

10. Line 480: I see now that engineering designs are not explicitly evaluated in this paper. My earlier comment (comment 3), asked about what engineering controls meant. The focus on engineering controls in the introduction section led me to believe that this paper would be about engineering controls. Rather it seems that this paper has implications for where to locate engineering controls but does not directly investigate this placement. If this is accurate, then I would suggest deemphasizing engineering controls from the introduction section.

This is correct that we use siting of engineering controls as a motivating reason for doing a spatially distributed sensitivity analysis. Calibrated models are used to optimize these and other water management decisions, and parameter screening is used to reduce the

dimensionality of the search to make the calibration more tractable. We propose broadening the introduction to say "engineering controls and water management decisions" to be more applicable. We changed the first sentence of the introduction to:

Spatially distributed hydrologic models are commonly employed to inform water management decisions across a watershed, such as the optimization of locations of engineering control measures.

11. From what I understood of this article, the first main finding was that parameters describing watershed characteristics are sometimes important for modeling hillslope hydrologic response even though they do not affect the streamflow at the model outlet much. The authors state that this might be important in the spatial location of engineering controls. There are many other reasons why getting the hydrology right within the watershed is important (modeling of spatially distributed soil moisture, etc.), but a major limitation is that we don't normally have data to compare to within the watershed, so in practice it would be hard to calibrate these parameters that don't affect streamflow much. The second main finding was that commonly used metrics (e.g., NSE) are not as sensitive to the decision relevant streamflows that we would want them to be. These are both important findings and points to make, but I found the article overall hard to read and understand. The authors may be served by focusing the text on the main findings and reducing discussion of peripheral topics.

These are two key points of the article. Because there are so few papers on decision-relevant sensitivity metrics, we thought it would be useful to provide an extended discussion that describes the importance of having the decision objectives in mind when completing a sensitivity analysis and subsequent calibration and optimization. Spatial sensitivity analysis is also often limited by data in sub-catchments and can lead to calibration challenges. We think it is important to discuss how the resulting parametric uncertainty can be used in robust optimizations.

We propose a better framing of the intent to use SA results to inform calibration of a model that is used to optimize decisions. We propose adding this sentence to the end of the first paragraph (same sentence as mentioned in reply to comment 5):

In this paper, we evaluate the influence of decision-relevant and calibration-relevant sensitivity metrics on parameter selection for calibration, and discuss the potential implications on subsequent model calibration and optimization of water management decisions.

---

## Referee Report (RR1)

**Review of: "Guidance on evaluating parametric model uncertainty at decision-relevant scales"**

*Summary and recommendation*

This paper aims to evaluate the variability in model parameters that result from sensitivity analyses based on (1) different spatial scales; and (2) different objective functions/ metrics. The paper is well written overall, and I think makes very salient and clear arguments for the importance of considering spatial scales and metrics relevant for management. I appreciate that the authors clearly put a lot of really hard work into this research, and think that the message is important for the larger scientific community. However, I think that the important message that this research is trying to get across is getting lost in the details, some of which are only peripherally related to the research. I think that the authors can do more to clarify the message, and remove non-essential points (or move to the SI). I further found some of the figures difficult to decipher, and the framework in the introduction a little confusing/ jumbled. Overall I believe that this paper will be a fine contribution to HESS once these issues are addressed.

*Major comments*

1. The text is dense and overly detailed in some places. I believe the take home messages are important but are getting lost in the details. I suggest the authors remove any methods, results, and discussion section that isn't relevant to the main points to the SI. For example:
   a. I think the entire section 2.3.1 (Elementary Effects for Parameters with Relational Constraints) is not essential to have in the main text. Rather, the authors could briefly state that the 271 parameters in RHYSSes were summarized into 237 parameters by combining those that are structurally dependent (and then refer to the SI for additional detail).
   b. Since the authors did not conduct a stochastic modeling approach, leave any discussion related to stochastic modeling (i.e., much of Section 2.1) to the discussion.

2. Terminology should be simplified throughout.
   a. Objectives vs. sensitivity metrics vs. objective functions vs. performance measure. These terms are used throughout and should be clarified. The term "objectives" is particularly confusing as the reader may associate this term with "objective function", which I believe the authors refer to as "sensitivity metrics" (?). I suggest that the authors read through the MS carefully and think about where it is possible to simplify and reduce these terms.
      i. For example, lines 130 – 135, the authors use "objectives" to refer to overarching management goals for water quality and quantity ("We consider water quantity and quality objectives as they are among the most common for hydrological modeling studies"), then sensitivity metrics to refer to flooding, low flow, and other flow objectives ("We evaluate three streamflow sensitivity metrics relevant to flooding, low flow, and all other flow objectives, respectively"), and then define these again as objectives ("These mutually exclusive objectives are respectively quantified as 1) flows greater than the historical 95th percentile, 2) flows less than the historical 5th percentile, and 3) flows between the historical 5th and

95th percentiles"). This is only one example of excessive/ confusing use of these terms.

ii. It seems that there is a lot of overlap between these terms, or maybe they are the same. In any case, I suggest creating a table that describes the different levels and defines the metric names, and then use these metric names consistently throughout, see example below.

| Applied to | | Scale | Performance measure | Sensitivity metric name |
|---|---|---|---|---|
| *Decision - relevant metrics* | | | | |
| Flow | High flows | Basin | SAE | Basin $_{high\ flows}$ |
| | | Hillslope | SAMD | Hillslope $_{high\ flows}$ |
| | Low flows | Basin | SAE | Basin $_{low\ flows}$ |
| | | Hillslope | SAMD | Hillslope $_{low\ flows}$ |
| | Other flows | Basin | MAE SAE | Basin $_{other\ flows}$ |
| | | Hillslope | SAMD | Hillslope $_{other\ flows}$ |
| Water quality | High TN concentration | Basin | SAE | TN $_{high}$ |
| | Low TN concentration | Basin | SAE | TN $_{low}$ |
| | Mean TN concentration | Basin | SAE | TN $_{mean}$ |
| *Calibration – relevant metrics* | | | | |
| Flow | All flows | Basin | NSE | ? |
| | | | LNSE | ? |
| | | | pBias | ? |
| | | | Log likely-hood | ? |

a. The 95th percentile terms are confusing on Figure 1; I was getting the lines on the plot confused with the 95th percentile flow sensitivity metrics, 5th percentile flows sensitivity metrics, etc. I think the authors could simplify and clarify by referring to the 95th percentile flows simply as "high flows", and 5th percentile flows as "low flows" after defining in table.

3. Introduction could be improved to better frame the two issues this paper is tackling. From my understanding, the main two issues are: (1) spatial scales of calibration do not match spatial scales relevant for management; and (2) calibration performance metrics do not represent hydrologic outcomes relevant for management. I think the issue of equifinality – which is relevant for issue (1) -- is getting mixed up in issue (2) in lines 38 – 52. Below I suggest an outline for first few paragraphs of the introduction.

   a. Management controls are spatially distributed throughout a watershed, and therefore modeling management approaches often call for spatially explicit models (i.e., distributed models)

      i. Distributed models require calibration of many parameters, some of which are not even observable

      ii. This calibration is challenging since observations are rarely available at scales needed to constrain all of these parameters; watershed outlets are gauged only so calibration is performed at the watershed scale.

        iii.  This leads to the well-known issue of equifinality when unknown parameter values are not constrained→ many ways to get to the same answer at the outlet→ large uncertainty in parameter values at local (finer) scales.

    b.  This presents two major challenges for modeling studies that aim to evaluate impacts of decision making….

        i.  Spatial scales of calibration do not match spatial scales relevant for management. Equifinality is particularly problematic for watershed models that aim to predict effects or optimize locations of management controls, since these are sensitive to local scale parametrization (which is highly uncertain when the model is only calibrated to a single location)

        ii.  Calibration performance metrics do not represent hydrologic outcomes relevant for management. A further, even more basic issue faced by modeling management decisions is that the majority of calibration performance metrics (e.g., NSE) are not necessarily, or explicitly, sensitive to hydrologic outcomes relevant for decision making (e.g., high flows, low flows).

    c.  The combination of these two issues have consequences….

4.  Figures should be simplified and made more legible. I provide detailed suggestions below.

5.  If the authors found similar parameter selections using SAMD and SAE at the basin scale (lines 120 – 124), why did they proceed with SAE for the basin outlet scale? It seems like for the purposes of comparing hillslope scale to basin scale selected parameters, it would be more defensible to use SAMD for both.

*Minor comments*

- Many opportunities to simplify language by re-arranging sentences and avoiding passive voice, for example:
  - Line 23: "such as the optimization of locations of engineering control measures…" could be simplified to "such as the optimal locations of engineering control measures…"
  - Line 24: "Accurate simulations of streamflows and nutrient fluxes in ungauged locations are desired to estimate the impact of control measures…" could be simplified to "Quantifying the impact of control measures requires accurate estimates of streamflow and nutrient fluxes in ungauged locations…"
- Line 25: "…on multiple objective functions" This is not essential to the point this paragraph is trying to make, and also introduces a new term that hasn't been defined. If the authors think that mentioning multiple objective functions is necessary in the second sentence of the MS, I suggest defining it first. Otherwise, remove from this sentence.
- Line 30 – 32: "Reviews of sensitivity …. at the outset of a study." These two sentences could be shortened and simplified: "Recent reviews of sensitivity analysis methods for spatially distributed models (e.g., Pianosi et al., 2016; Razavi and Gupta, 2015; Koo et al., 2020b; Lilburne and Tarantola, 2009) emphasize the need to consider, at the outset of a study, the definition of sensitivity within the study context."
- Line 33: "decision objective values" is a confusing term that has not been defined yet. What are "decision objectives" and how are they different from "decision objective values" in this sentence?

- Line 35-37: "In this study, we evaluate…water management decisions". This sentence seems like it would fit better towards the end of the introduction – I was a little thrown off that the authors describe the objectives of the paper that at the end of the first paragraph, but then go on to provide further motivation/ background (P2), and then go back to the objectives of the paper again (P3).
- Line 50-51: "This would suggest there is equifinality…across the watershed." I'm not sure that the fact that distributed stormwater control outcomes are affected by different parameters than watershed scale outcomes suggests that there is equifinality. Equifinality exists regardless of whether a stormwater control is being simulated in the model. I think I would suggest the authors use the fact that equifinality is a rampant issue in distributed models and poses unique challenges for simulating stormwater control measures, which are often distributed across a watershed. In other words, introduce equifinality earlier on in the introduction (i.e., in P1 where the authors describe the fact that these models have hundreds of parameters that need to be calibrated).
- Line 78-72: "the results we obtain…impact on sensitivity metrics." These two sentences are confusing as they are written in a passive voice; it is unclear whether the authors "provide general guidelines for spatially distributed models" and "inform prioritization of data collection efforts", or whether this was done separately/ by another study/ in practice.
- Line 93-95: "If employing a stochastic modeling approach…could be considered in a sensitivity analysis". again, since this paper focuses on parametric uncertainty and assumes a static model, this does not seems relevant and could be removed. Moreover, these lines include terms that are not (a) defined previously, like error model shape, and (b) are not used again in the manuscript – this additional information detracts from the main point of the paper by distracting the reader (or, at least me!).
- Line 118 – 119: "Because performance measures require an observation time series to compute, we needed a different approach to measure relative variability for hillslope sensitivity analysis. At the hillslope scale, we use…" I suggest rephrasing and simplifying: "At the hillslope scale (where observation time series are not available), we use the sum of absolute median deviation…"
- Line 130 – 133: "We consider water quantity and quality objectives …. historical 5th and 95th percentiles." These sentences are a little confusing because there are so many different terms used and it's not clear what they all refer to (see major comment 2a above). Suggested revision: "We consider sensitivity metrics related to decision-making for water quantity and quality outcomes as they are among the most common for hydrological modeling studies. For water quality, we quantify SAE (basin scale) and SAMD (hillslope scale) separately for (1) high flows (flows greater than the historical 95th percentile), (2) low flows (flows less than the historical 5th percentile), and (3) all other flows (flows between the historical 5th and 95th percentiles)."
- Lines 143 – 145: Somewhere in here the authors should state which performance measure they used here (SAE?).
- Lines 165 – 167: "We selected the likelihood model based on…which is a generalized normal distribution." Suggest simplifying: "We selected the skew exponential power model (a generalized normal distribution) as the likelihood model due to its ability to fit the wide range of residual distribution shapes that result from random sampling."
- Line 237: "Then, we flagged…" Does "flagged" mean "selected"?
- Lines 269 – 272: "While authors Lin and Band…unrealistic mortality)." This sentence isn't essential for the point of the paragraph. I suggest moving this to the discussion or SI.
- Section 4. Case study site description. The order of the sentences in this paragraph are a little disjointed. I suggest moving lines 341 – 344 ("The Baisman Run watershed…reforestation optimization.") to before the sentence starting on line 337 ("After a five year spin-up period…").

This would make it so first you present all of the background info on the watershed, and then you discuss your modeling approach. As it is, you describe the watershed, discuss your modeling approach, and then describe the watershed again.

- Line 334 – 345: "The goal of this sensitivity analysis is to inform the selection of parameters to calibrate a RHESSys model that could be used in such a reforestation optimization." This was surprising to me, since the introduction really focused on stormwater control measures, not reforestation. If this truly is the goal of the paper, the introduction needs to be revised to focus on reforestation efforts. Also, this is a strange place to put the goal of the paper – it should be in the introduction (and it is, in fact, but the introduction states that "The goal is to discover to which parameters the decision objectives are most sensitive across the watershed", which is different than that stated in lines 334 – 345).
- Lines 301 – 307: This paragraph might fit better at the end of a section (i.e., end of the intro, methods or case study site description).
- Lines 369 – 271: If I am interpreting this correctly, these lines are saying that 21 parameters were selected for basin outlet, 18 of which were based on streamflow metrics, and 19 based on TN metrics. This would imply that out of the 21 parameters selected, only 5 are not overlapping between the streamflow and TN metrics. This, to me, does not necessarily support "using sensitivity metrics for each output variable or objective" since there is actually a lot of overlap between the parameters that were selected.
- Line 375: top row should be left column
- Line 393: bottom row should be right column
- Line 409 – 411: "The majority of the watershed is forested…correspond to power lines." This seems like watershed background that should be moved to the case study site description (Section 4)
- Line 581 – 582: "If we select all parameters…that will lead to more parameters compared to using only the gauge location." This sentence is confusing, suggest revising: "More calibration parameters result from sensitivity analysis at local scales (i.e. ungauged hillslope) than do from sensitivity analysis at watershed scales."

*Figure comments*

- Suggest adding a conceptual figure to the beginning of the methods to describe overall approach
- Figure 2.
  - Suggest transposing the subplots so that the flow metrics are all along a single row, and TN metrics are in the second row. This would make it easier to compare across the different flow and TN metrics.
  - Suggest only showing those that meet the 10% threshold (very hard to distinguish between lines as is, lots of the numbers overlap)
    - This could free up some space along the x-axis for parameter names, rather than symbols/ numbers
  - The caption says this provides the EEs for "the six sensitivity metrics", but I only see SAE, which would imply this is only for the basin scale decision-relevant metrics? What about SAMD (hillslope scale), and all calibration relevant metrics?  The text (line 372) says Figure 2 shows "basin scale EEs", but still this doesn't explain why calibration relevant metrics aren't included.  Again, I think this is an issue of terminology and should be

clarified throughout, but I point it out specifically here since the caption of the figure is incorrect, or the text is misleading.

- Figure 3
    - Separate into two figures: one with land cover maps and hillslopes (currently A and B), and one with EE ranks and indicators (currently C and D).
    - Make the land cover maps Figure 1, move up to be with the case study site description (Section 4), where they are already referenced
    - To further simplify this figure, consider grouping the hillslopes based on relevant properties (i.e, forested/ non-forested/ impervious) and using the mean EE across hillslopes in that group. This would be more meaningful for the reader (and would support the points the authors make in lines 412 – 439), and would simplify the figure a lot.

---

## Author Response (AR2)

**Authors' Response to Editor and Reviewers of "Guidance on evaluating parametric model uncertainty at decision-relevant scales" by Smith et al.**

Editor and Reviewer comments are in black. Line numbers in our replies correspond to the revised manuscript PDF without tracked changes in it.

Author responses are in blue
Text edits are in red

**Editor**

Dear authors,
I have now received three anonymous reviews in response to your revised article. One reviewer is happy with the changes you've made, while two other reviewers encourage further revision of your article. The majority of the reviewers were positive and commented on the strong substance of your article, but encourage some reframing of your article (particularly in the introduction). The reviewers also noted a few minor changes and clarifications needed as well.

Overall, I encourage you to undertake the recommendations provided by the reviewers. I look forward to receiving your revised manuscript.

We have revised the manuscript introduction and made other clarifying suggestions to the text. We also revised the figures and added a summary table, as suggested by Reviewer 4.

**Reviewer #1**

The authors have done a good job addressing my comments and I do not have remaining comments.

Thank you for reviewing our revised manuscript. Your comments in the first round greatly improved the paper.

**Reviewer #3**

The idea behind the manuscript is interesting however, I find that the overall organization of the paper should be improved to favor the comprehension of researchers not familiar with GSA methods. Diverse parts of the manuscript are difficult to read and the quality of the figures still needs to be improved.

As a second major point, I invite the authors to revise the literature carefully because there are some relevant studies on sensitivity-based parameter calibration, for the identification of the relative importance of the parameters, and the locations where measurements should be

collected being the model most sensitive to its parameters.

If the authors will address the two major points above, the novelty that the paper brings will be clarified as well as the possibility to replicate the methodology will be facilitated.

Thank you for reviewing our manuscript.

For your first point, we cite many recent review papers on GSA and spatial SA. Because these review articles are available, we do not see a need to be more introductory to GSA methods in this paper.

We also modified the figures and added a summary table, as discussed in our response to Reviewer 4.

For your second point, the purpose of this paper is to provide decision-relevant guidance, not general guidance, on sensitivity-based parameter calibration. We agree there are many studies that provide general guidance, and we already cite some examples in the manuscript. However, we are not aware of studies that focus on a decision-informed SA for model calibration.

**Reviewer #4**

This paper aims to evaluate the variability in model parameters that result from sensitivity analyses based on (1) different spatial scales; and (2) different objective functions/ metrics. The paper is well written overall, and I think makes very salient and clear arguments for the importance of considering spatial scales and metrics relevant for management. I appreciate that the authors clearly put a lot of really hard work into this research, and think that the message is important for the larger scientific community. However, I think that the important message that this research is trying to get across is getting lost in the details, some of which are only peripherally related to the research. I think that the authors can do more to clarify the message, and remove non-essential points (or move to the SI). I further found some of the figures difficult to decipher, and the framework in the introduction a little confusing/ jumbled. Overall I believe that this paper will be a fine contribution to HESS once these issues are addressed.

Thank you for thoroughly reviewing our manuscript. We think your suggestions help to improve the clarity of our work. Please see our responses to your comments below.

*Major comments*
1. The text is dense and overly detailed in some places. I believe the take home messages are important but are getting lost in the details. I suggest the authors remove any methods, results, and discussion section that isn't relevant to the main points to the SI.

We have shortened and merged some sentences and paragraphs in the revised version. Please see the tracked changes for these minor edits.

For example:

a. I think the entire section 2.3.1 (Elementary Effects for Parameters with Relational Constraints) is not essential to have in the main text. Rather, the authors could briefly state that the 271 parameters in RHYSSes were summarized into 237 parameters by combining those that are structurally dependent (and then refer to the SI for additional detail).

> We accepted this suggestion and added the following sentence in the Section 3 model description.
>
> Some parameters are structurally dependent, so we aggregated EEs for these parameters, resulting in 237 unique EEs for each sensitivity metric (supplementary information item S0 describes the aggregation method)

b. Since the authors did not conduct a stochastic modeling approach, leave any discussion related to stochastic modeling (i.e., much of Section 2.1) to the discussion.

> We mention stochastic modeling in Section 2.1 because it motivates considering model error for TN. We think that the uncertainty sources associated with an error model make more sense in Section 2.1 where other uncertainty sources are described.

2. Terminology should be simplified throughout.

a. Objectives vs. sensitivity metrics vs. objective functions vs. performance measure. These terms are used throughout and should be clarified. The term "objectives" is particularly confusing as the reader may associate this term with "objective function", which I believe the authors refer to as "sensitivity metrics" (?). I suggest that the authors read through the MS carefully and think about where it is possible to simplify and reduce these terms.

i. For example, lines 130 – 135, the authors use "objectives" to refer to overarching management goals for water quality and quantity ("We consider water quantity and quality objectives as they are among the most common for hydrological modeling studies"), then sensitivity metrics to refer to flooding, low flow, and other flow objectives ("We evaluate three streamflow sensitivity metrics relevant to flooding, low flow, and all other flow objectives, respectively"), and then define these again as objectives ("These mutually exclusive objectives are respectively quantified as 1) flows greater than the historical 95th percentile, 2) flows less than the historical 5th percentile, and 3) flows between the historical 5th and 95th percentiles"). This is only one example of excessive/ confusing use of these terms.

ii. It seems that there is a lot of overlap between these terms, or maybe they are the same. In any case, I suggest creating a table that describes the different levels and defines the metric names, and then use these metric names consistently throughout, see example below.

| Applied to | | Scale | Performance measure | Sensitivity metric name |
|---|---|---|---|---|
| *Decision - relevant metrics* | | | | |
| Flow | High flows | Basin | SAE | Basin $_{high\ flows}$ |
| | | Hillslope | SAMD | Hillslope $_{high\ flows}$ |
| | Low flows | Basin | SAE | Basin $_{low\ flows}$ |
| | | Hillslope | SAMD | Hillslope $_{low\ flows}$ |
| | Other flows | Basin | MAE SAE | Basin $_{other\ flows}$ |
| | | Hillslope | SAMD | Hillslope $_{other\ flows}$ |
| Water quality | High TN concentration | Basin | SAE | TN $_{high}$ |
| | Low TN concentration | Basin | SAE | TN $_{low}$ |
| | Mean TN concentration | Basin | SAE | TN $_{mean}$ |
| *Calibration – relevant metrics* | | | | |
| Flow | All flows | Basin | NSE | ? |
| | | | LNSE | ? |
| | | | pBias | ? |
| | | | Log likely-hood | ? |

Thank you for pointing out the potential for confusion among these terms. They are all different, with the exception of "sensitivity metrics" and "objective function", the latter of which we have removed from the paper to avoid confusion. We already had a paragraph dedicated to the definition of sensitivity metrics vs. performance measures (Section 2.2). The definition of objective is defined in the introduction for calibration and decision objectives. In most cases, we have rephased as "decision maker's objectives" to be clear this is not an objective function.

We say in Section 2.2.2 that the performance measure is as you list in the table and the sensitivity metric is the application of the performance measure to all flows. We do not use MAE for other flows or anywhere in the manuscript.

For the first use of performance measure, we now distinguish it from sensitivity metric:

Common calibration performance measures are employed as sensitivity metrics by evaluating performance across all flow magnitudes

We have also added a new table in line with what you suggested:

**Table 1.** Table of decision-relevant and calibration-relevant sensitivity metrics.

| Sensitivity Metric | | Scale | Performance Measure |
|---|---|---|---|
| **Decision-Relevant Metrics** | | | |
| Streamflow | High Flow Days | Basin | SAE |
| | Low Flow Days | | |
| | Other Days | | |
| | High Flow Days | Hillslope | SAMD |
| | Low Flow Days | | |
| | Other Days | | |
| TN Concentration | High TN, All Days | Basin | SAE |
| | Mean TN, All Days | | |
| | Low TN, All Days | | |
| **Calibration-Relevant Metrics** | | | |
| Streamflow | All Flows, All Days | Basin | NSE |
| | | | LNSE |
| | | | pBias |
| | | | LogL |

a. The 95th percentile terms are confusing on Figure 1; I was getting the lines on the plot confused with the 95th percentile flow sensitivity metrics, 5th percentile flows sensitivity metrics, etc. I think the authors could simplify and clarify by referring to the 95th percentile flows simply as "high flows", and 5th percentile flows as "low flows" after defining in table.

We edited the legend title to say "Bootstrapped EE value: Scale"

[Figure]

3. Introduction could be improved to better frame the two issues this paper is tackling. From my understanding, the main two issues are: (1) spatial scales of calibration do not match spatial scales relevant for management; and (2) calibration performance metrics do not represent hydrologic outcomes relevant for management. I think the issue of equifinality – which is relevant for issue (1) -- is getting mixed up in issue (2) in lines 38 – 52. Below I suggest an outline for first few paragraphs of the introduction.

    a. Management controls are spatially distributed throughout a watershed, and therefore modeling management approaches often call for spatially explicit models (i.e., distributed models)

        i. Distributed models require calibration of many parameters, some of which are not even observable

        ii. This calibration is challenging since observations are rarely available at scales needed to constrain all of these parameters; watershed outlets are gauged only so calibration is performed at the watershed scale.

        iii. This leads to the well-known issue of equifinality when unknown parameter values are not constrained⯑ many ways to get to the same answer at the outlet⯑ large uncertainty in parameter values at local (finer) scales.

    b. This presents two major challenges for modeling studies that aim to evaluate impacts of decision making….

        i. Spatial scales of calibration do not match spatial scales relevant for management. Equifinality is particularly problematic for watershed

models that aim to predict effects or optimize locations of management controls, since these are sensitive to local scale parametrization (which is highly uncertain when the model is only calibrated to a single location)

    ii. Calibration performance metrics do not represent hydrologic outcomes relevant for management. A further, even more basic issue faced by modeling management decisions is that the majority of calibration performance metrics (e.g., NSE) are not necessarily, or explicitly, sensitive to hydrologic outcomes relevant for decision making (e.g., high flows, low flows).

    c. The combination of these two issues have consequences….

Thank you for this suggested reorganization of the introduction. We have taken your suggestion to restructure the first 2 paragraphs of the introduction by presenting equifinality in the first paragraph.

4. Figures should be simplified and made more legible. I provide detailed suggestions below.

Thanks for these suggestions. See our comments below for how we addressed them.

5. If the authors found similar parameter selections using SAMD and SAE at the basin scale (lines 120 – 124), why did they proceed with SAE for the basin outlet scale? It seems like for the purposes of comparing hillslope scale to basin scale selected parameters, it would be more defensible to use SAMD for both.

We were asked in the first revision to evaluate SAMD for the basin outlet and found that the results were similar. Keeping SAE in the main text was largely a time consideration. We point to the supplementary material for a comparison of SAMD and SAE for decision-relevant flow sensitivity metrics.

*Minor comments*

- Many opportunities to simplify language by re-arranging sentences and avoiding passive voice, for example:
  - Line 23: "such as the optimization of locations of engineering control measures…" could be simplified to "such as the optimal locations of engineering control measures…"
  - Line 24: "Accurate simulations of streamflows and nutrient fluxes in ungauged locations are desired to estimate the impact of control measures…" could be simplified to "Quantifying the impact of control measures requires accurate estimates of streamflow and nutrient fluxes in ungauged locations…"

We accepted these edits and simplified sentences in other parts of the manuscript.

- Line 25: "…on multiple objective functions" This is not essential to the point this paragraph is trying to make, and also introduces a new term that hasn't been defined. If the authors think that mentioning multiple objective functions is necessary in the second sentence of the MS, I suggest defining it first. Otherwise, remove from this sentence.

  We removed objectives from this sentence.

- Line 30 – 32: "Reviews of sensitivity …. at the outset of a study." These two sentences could be shortened and simplified: "Recent reviews of sensitivity analysis methods for spatially distributed models (e.g., Pianosi et al., 2016; Razavi and Gupta, 2015; Koo et al., 2020b; Lilburne and Tarantola, 2009) emphasize the need to consider, at the outset of a study, the definition of sensitivity within the study context."

  We simplified these sentences in line with your suggestion.

  Recent reviews of sensitivity analysis methods for spatially distributed models \citep{Pianosi2016,Razavi2015,Koo2020position,Lilburne2009} emphasize the critical need to answer, at the outset of a study, ``What is the intended definition for sensitivity in the current context?'' \citep{Razavi2015}.

- Line 33: "decision objective values" is a confusing term that has not been defined yet. What are "decision objectives" and how are they different from "decision objective values" in this sentence?

  We changed this sentence to be more general:

  For studies that aim to use the resulting model to spatially optimize decisions, sensitivity should be defined for the objectives of the decision maker.

- Line 35-37: "In this study, we evaluate…water management decisions". This sentence seems like it would fit better towards the end of the introduction – I was a little thrown off that the authors describe the objectives of the paper that at the end of the first paragraph, but then go on to provide further motivation/ background (P2), and then go back to the objectives of the paper again (P3).

  We have removed this sentence because we have a similar sentence later in the introduction.

- Line 50-51: "This would suggest there is equifinality…across the watershed." I'm not sure that the fact that distributed stormwater control outcomes are affected by different parameters than watershed scale outcomes suggests that there is equifinality. Equifinality exists regardless of whether a stormwater control is being simulated in the model. I think I would suggest the authors use the fact that equifinality is a rampant issue in distributed models and poses unique challenges for simulating stormwater control measures, which are often distributed across a watershed. In other words, introduce equifinality earlier on in the introduction (i.e., in P1 where the authors

describe the fact that these models have hundreds of parameters that need to be calibrated).

> We edited the introduction to introduce the concept of equifinality in paragraph 1.

- Line 78-72: "the results we obtain...impact on sensitivity metrics." These two sentences are confusing as they are written in a passive voice; it is unclear whether the authors "provide general guidelines for spatially distributed models" and "inform prioritization of data collection efforts", or whether this was done separately/ by another study/ in practice.

> We have rewritten to say:
>
> We use the results of a comprehensive sensitivity analysis of all non-structural model parameters to provide general guidelines for spatially distributed models and some specific recommendations for RHESSys users.

- Line 93-95: "If employing a stochastic modeling approach...could be considered in a sensitivity analysis". again, since this paper focuses on parametric uncertainty and assumes a static model, this does not seems relevant and could be removed. Moreover, these lines include terms that are not (a) defined previously, like error model shape, and (b) are not used again in the manuscript – this additional information detracts from the main point of the paper by distracting the reader (or, at least me!).

> The TN model considers residual error, and we think this paragraph helps to motivate considering residual error in SA. We provide citations to papers that explore each of the concepts in more detail. We revised the error model sentence to say:
>
> ...additional uncertainty sources include the choice of residual error model shape (e.g., lognormal) \citep{Smith2015}...

- Line 118 – 119: "Because performance measures require an observation time series to compute, we needed a different approach to measure relative variability for hillslope sensitivity analysis. At the hillslope scale, we use..." I suggest rephrasing and simplifying: "At the hillslope scale (where observation time series are not available), we use the sum of absolute median deviation..."

> We edited this sentence:
>
> For hillslopes (where observations are not available) we used the sum of absolute median deviation (SAMD), where the median value for each hillslope was computed across all model simulations.

- Line 130 – 133: "We consider water quantity and quality objectives …. historical 5th and 95th percentiles." These sentences are a little confusing because there are so many different terms used and it's not clear what they all refer to (see major comment 2a above). Suggested revision: "We consider sensitivity metrics related to decision-making for water quantity and quality outcomes as they are among the most common for hydrological modeling studies. For water quality, we quantify SAE (basin scale) and SAMD (hillslope scale) separately for (1) high flows (flows greater than the historical 95th percentile), (2) low flows (flows less than the historical 5th percentile), and (3) all other flows (flows between the historical 5th and 95th percentiles)."

  We simplified these sentences based on your suggestion.

  We consider sensitivity metrics that are relevant to water quantity and quality outcomes because they are among the most common for hydrological modeling studies. For water quantity, we compute SAE (basin) and SAMD (hillslopes) for three mutually exclusive flows: 1) high flows greater than the historical $95^{th}$ percentile, 2) low flows less than the historical $5^{th}$ percentile, and 3) all other flows between the historical $5^{th}$ and $95^{th}$ percentiles.

- Lines 143 – 145: Somewhere in here the authors should state which performance measure they used here (SAE?).

  Added.

  The water quality sensitivity metrics are the SAE for…

- Lines 165 – 167: "We selected the likelihood model based on…which is a generalized normal distribution." Suggest simplifying: "We selected the skew exponential power model (a generalized normal distribution) as the likelihood model due to its ability to fit the wide range of residual distribution shapes that result from random sampling."

  We edited this sentence:

  We selected the skew exponential power (generalized normal) distribution \citep{Schoups2010} as the likelihood model due to its ability to fit a wide variety of residual distribution shapes that could result from random sampling of hydrological model parameters.

- Line 237: "Then, we flagged…" Does "flagged" mean "selected"?

  We revised this sentence to remove flagged.

  All of the parameters whose estimated $95^{th}$ percentile EE values were greater than this cutoff value would be selected for calibration for that metric.

- Lines 269 – 272: "While authors Lin and Band…unrealistic mortality)." This sentence isn't essential for the point of the paragraph. I suggest moving this to the discussion or SI.

  We think it's important to keep part of this sentence to justify why we didn't simulate nitrogen from RHESSys.

  We found that randomly sampling non-structural growth model parameters within their specified ranges commonly resulted in unstable ecosystems (e.g., very large trees or unrealistic mortality).

- Section 4. Case study site description. The order of the sentences in this paragraph are a little disjointed. I suggest moving lines 341 – 344 ("The Baisman Run watershed…reforestation optimization.") to before the sentence starting on line 337 ("After a five year spin-up period…"). This would make it so first you present all of the background info on the watershed, and then you discuss your modeling approach. As it is, you describe the watershed, discuss your modeling approach, and then describe the watershed again.

  We accepted this suggestion.

- Line 334 – 345: "The goal of this sensitivity analysis is to inform the selection of parameters to calibrate a RHESSys model that could be used in such a reforestation optimization." This was surprising to me, since the introduction really focused on stormwater control measures, not reforestation. If this truly is the goal of the paper, the introduction needs to be revised to focus on reforestation efforts. Also, this is a strange place to put the goal of the paper – it should be in the introduction (and it is, in fact, but the introduction states that "The goal is to discover to which parameters the decision objectives are most sensitive across the watershed", which is different than that stated in lines 334 – 345).

  Thanks for pointing out that we say reforestation here. An in-prep paper based on this work focuses on reforestation. We have changed this to say "stormwater infrastructure optimization" in this paragraph. We also edited the short non-technical summary of our paper

  Watershed models are used to simulate streamflow and water quality, and to inform siting and sizing decisions for runoff and nutrient control projects. Data are limited for many watershed processes that are represented in such models, which requires selecting the most important processes to be calibrated. We show that this selection should be based on decision-relevant metrics at the spatial scales of interest for the control projects. This should enable more robust project designs.

  The use of "goal" here is a poor word choice. We replaced "goal" with "hypothetical motivation" so it's not confused with the goal of our paper.

The hypothetical motivation for this sensitivity analysis is to inform the selection of parameters to calibrate a RHESSys model that could be used in an optimization of stormwater infrastructure that aims to control flooding and reduce nutrient exports.

- Lines 301 – 307: This paragraph might fit better at the end of a section (i.e., end of the intro, methods or case study site description).

    The paragraph is currently at the end of the section describing the RHESSys model, and that seems like the best place to us. We do not want to end the Introduction or Methods with this paragraph, as it describes a less general contribution of our work to understanding parameter sensitivities in RHESSys, specifically; however, we do not want to end the Case Study Site Description with this paragraph, as it describes a more general contribution than to just our model site.

- Lines 369 – 271: If I am interpreting this correctly, these lines are saying that 21 parameters were selected for basin outlet, 18 of which were based on streamflow metrics, and 19 based on TN metrics. This would imply that out of the 21 parameters selected, only 5 are not overlapping between the streamflow and TN metrics. This, to me, does not necessarily support "using sensitivity metrics for each output variable or objective" since there is actually a lot of overlap between the parameters that were selected.

    You're interpreting correctly. If a set of sensitivity metrics for streamflow or TN were used, then there would be either 2 or 3 parameters missing from the calibration that are statistically significantly important for the other set of metrics. Choosing to not include them wouldn't be justifiable. While this may seem like a small number of parameters to miss, excluding them could have decision-relevant implications for stormwater design. We would also like to note that it is somewhat surprising the parameters are not fully overlapping since our TN modeling is based on a regression with streamflow.

- Line 375: top row should be left column

- Line 393: bottom row should be right column
    We made these edits.

- Line 409 – 411: "The majority of the watershed is forested…correspond to power lines." This seems like watershed background that should be moved to the case study site description (Section 4)
    We moved these sentences to Section 4.

- Line 581 – 582: "If we select all parameters…that will lead to more parameters compared to using only the gauge location." This sentence is confusing, suggest revising: "More calibration parameters result from sensitivity analysis at local scales (i.e. ungauged hillslope) than do from sensitivity analysis at watershed scales."

  We accepted this edit.

*Figure comments*
- Suggest adding a conceptual figure to the beginning of the methods to describe overall approach

  We added the table that you suggested and we think that serves well as a conceptual overview of what sensitivity metrics and scales we compare.

- Figure 2.
  - Suggest transposing the subplots so that the flow metrics are all along a single row, and TN metrics are in the second row. This would make it easier to compare across the different flow and TN metrics.

    We had the figure in this orientation in the previous version of the manuscript, but the figure panels were too small to be useful.
  - Suggest only showing those that meet the 10% threshold (very hard to distinguish between lines as is, lots of the numbers overlap)
    - This could free up some space along the x-axis for parameter names, rather than symbols/ numbers

      We edited the figure to show only those parameters that are selected by any of the metrics in Table 1. We kept the symbols because they are used in Figure 3

  - The caption says this provides the EEs for "the six sensitivity metrics", but I only see SAE, which would imply this is only for the basin scale decision-relevant metrics? What about SAMD (hillslope scale), and all calibration relevant metrics? The text (line 372) says Figure 2 shows "basin scale EEs", but still this doesn't explain why calibration relevant metrics aren't included. Again, I think this is an issue of terminology and should be clarified throughout, but I point it out specifically here since the caption of the figure is incorrect, or the text is misleading.

    We have edited the figure caption to:

    Mean absolute value of elementary effects for RHESSys model parameters evaluated for the six decision-relevant sensitivity metrics at the basin outlet.

[Figure]

- Figure 3
  - Separate into two figures: one with land cover maps and hillslopes (currently A and B), and one with EE ranks and indicators (currently C and D).
  - Make the land cover maps Figure 1, move up to be with the case study site description (Section 4), where they are already referenced
  - To further simplify this figure, consider grouping the hillslopes based on relevant properties (i.e, forested/ non-forested/ impervious) and using the mean EE across hillslopes in that group. This would be more meaningful for the reader

(and would support the points the authors make in lines 412 – 439), and would simplify the figure a lot.

Thank you for these suggestions. We have modified the figure in line with these suggestions. Panel A is now the former panel C, and panel B is the former panel D with an additional aggregation across hillslopes 1-8 and 9-14.